# An Improved Analysis of Stochastic Gradient Descent with Momentum

**Yanli Liu**[*]   **Yuan Gao**[†]   **Wotao Yin**[*]
[*]Department of Mathematics, University of California, Los Angeles
[†]Department of IEOR, Columbia University
{yanli, wotaoyin}@math.ucla.edu, {gao.yuan}@columbia.edu

## Abstract

SGD with momentum (SGDM) has been widely applied in many machine learning tasks, and it is often applied with dynamic stepsizes and momentum weights tuned in a stagewise manner. Despite of its empirical advantage over SGD, the role of momentum is still unclear in general since previous analyses on SGDM either provide worse convergence bounds than those of SGD, or assume Lipschitz or quadratic objectives, which fail to hold in practice. Furthermore, the role of dynamic parameters has not been addressed. In this work, we show that SGDM converges as fast as SGD for smooth objectives under both strongly convex and nonconvex settings. We also prove that multistage strategy is beneficial for SGDM compared to using fixed parameters. Finally, we verify these theoretical claims by numerical experiments.

## 1   Introduction

Stochastic gradient methods have been a widespread practice in machine learning. They aim to minimize the following empirical risk:

$$\min_{x \in \mathbb{R}^d} f(x) := \frac{1}{n} \sum_{i=1}^n \ell(x, q_i), \tag{1}$$

where $\ell$ is a loss function and $\{q_i\}_{i=1}^n$ denotes the training data, $x$ denotes the trainable parameters of the machine learning model, e.g., the weight matrices in a neural network.

In general, stochastic gradient methods can be written as

$$\begin{aligned} m^k &= \beta m^{k-1} + (1-\beta)\tilde{g}^k, \\ x^{k+1} &= x^k - \alpha m^k. \end{aligned} \tag{2}$$

where $\alpha > 0$ is a stepsize, $\beta \in [0, 1)$ is called momentum weight, and $m^0 = 0$. The classical Stochastic Gradient Descent(SGD) method [21] uses $\beta = 0$ and $m^k = \tilde{g}^k$, where $\tilde{g}^k$ is a stochastic gradient of $f(x)$ at $x^k$. To boost the practical performance, one often applies a momentum weight of $\beta > 0$. and the resulting algorithm is often called SGD with momentum (SGDM). SGDM is very popular for training neural networks with remarkable empirical successes, and has been implemented as the default SGD optimizer in Pytorch [19] and Tensorflow [1][1].

The idea behind SGDM originates from Polyak's heavy-ball method [20] for deterministic optimization. For strongly convex and smooth objectives, heavy-ball method enjoys an accelerated linear

convergence rate over gradient descent [7]. However, the theoretical understanding of its stochastic counterpart is far from being complete.

In the case of fixed stepsize and momentum weight, most of the current results only apply to restrictive settings. In [15, 16] and [12], the behavior of SGDM on least square regression is analyzed and linear convergence is established. [9] analyzes the local convergence rate of SGDM for strongly convex and smooth functions, where the initial point $x^0$ is assumed to be close enough to the minimizer $x^*$. [25] provides global convergence of SGDM, but only for objectives with *uniformly bounded gradients*, thus excluding many machine learning models such as Ridge regression. Very recently, [26] presents a convergence bound of $\mathcal{O}(\frac{1}{k\alpha} + \frac{\alpha}{1-\beta})$ for general smooth nonconvex objectives[3]. When $\beta = 0$, this recovers the classical convergence bound of $\mathcal{O}(\frac{1}{k\alpha} + \alpha)$ of SGD [4]. However, the size of stationary distribution $\mathcal{O}(\frac{\alpha}{1-\beta})$ is $\frac{1}{1-\beta}$ times larger than that of SGD. This factor is not negligible, especially when large $\beta$ values such as $0.99$ and $0.995$ is applied [24]. Therefore, their result does not explain the competitiveness of SGDM compared to SGD. Concurrent to this work, [22] shows that SGDM converges as fast as SGD under convexity and strong convexity, and that it is asymptotically faster than SGD for overparameterized models. Remarkably, their analysis considers a different stepsize and momentum weight schedule from this work, and applies to arbitrary sampling without assuming the bounded variance of the gradient noise.

In deep learning, SGDM is often applied with various parameter tuning rules to achieve efficient training. One of the most widely adopted rules is called "constant and drop", where a constant stepsize is applied for a long period and is dropped by some constant factor to allow for refined training, while the momentum weight is either kept unchanged (usually 0.9) or gradually increasing. We call this strategy Multistage SGDM and summarize it in Algorithm 1. Practically, (multistage) SGDM was successfully applied to training large-scale neural networks [13, 11], and it was found that appropriate parameter tuning leads to superior performance [24]. Since then, (multistage) SGDM has become increasingly popular [23].

At each stage, Multistage SGDM (Algorithm 1) requires three parameters: stepsize, momentum weight, and stage length. In [8] and [10], doubling argument based rules are analyzed for SGD on strongly convex objectives, where the stage length is doubled whenever the stepsize is halved. Recently, certain stepsize schedules are shown to yield faster convergence for SGD on nonconvex objectives satisfying growth conditions [27, 5], and a nearly optimal stepsize schedule is provided for SGD on least square regression [6]. These results consider only the momentum-free case. Another recent work focuses on the asymptotic convergence of SGDM (i.e., without convergence rate) [9], which requires the momentum weights to approach either 0 or 1, and therefore contradicts the common practice in neural network training. In summary, the convergence rate of Multistage SGDM (Algorithm 1) has not been established except for the momentum-free case, and the role of parameters in different stages is unclear.

---

**Algorithm 1** Multistage SGDM

---

**Input:** problem data $f(x)$ as in (1), number of stages $n$, momentum weights $\{\beta_i\}_{i=1}^{n} \subseteq [0,1)$, step sizes $\{\alpha_i\}_{i=1}^{n}$, and stage lengths $\{T_i\}_{i=1}^{n}$ at $n$ stages, initialization $x^1 \in \mathbb{R}^d$ and $m^0 = 0$, iteration counter $k = 1$.

1: **for** $i = 1, 2, ..., n$ **do**
2:      $\alpha \leftarrow \alpha_i, \beta \leftarrow \beta_i$;
3:      **for** $j = 1, 2, ..., T_i$ **do**
4:          Sample a minibatch $\zeta^k$ uniformly from the training data;
5:          $\tilde{g}^k \leftarrow \nabla_x l(x^k, \zeta^k)$;
6:          $m^k \leftarrow \beta m^{k-1} + (1-\beta)\tilde{g}^k$;
7:          $x^{k+1} \leftarrow x^k - \alpha m^k$;
8:          $k \leftarrow k + 1$;
9:      **end for**
10: **end for**
11: **return** $\tilde{x}$, which is generated by first choosing a stage $l \in \{1, 2, ...n\}$ uniformly at random, and then choosing $\tilde{x} \in \{x^{T_1+\cdots+T_{l-1}+1}, x^{T_1+\cdots+T_{l-1}+2}, ..., x^{T_1+\cdots+T_l}\}$ uniformly at random;

## 1.1 Our contributions

In this work, we provide new convergence analysis for SGDM and Multistage SGDM that resolve the aforementioned issues. A comparison of our results with prior work can be found in Table 1.

1. We show that for both strongly convex and nonconvex objectives, SGDM (2) enjoys the same convergence bound as SGD. This helps explain the empirical observations that SGDM is at least as fast as SGD [23]. Our analysis relies on a new observation that, the update direction $m^k$ of SGDM (2) has a controllable deviation from the current full gradient $\nabla f(x^k)$, and enjoys a smaller variance. Inspired by this, we construct a new Lyapunov function that properly handles this deviation and exploits an auxiliary sequence to take advantage of the reduced variance.

   Compared to aforementioned previous work, our analysis applies to not only least squares, does not assume uniformly bounded gradient, and improves the convergence bound.

2. For the more popular SGDM in the multistage setting (Algorithm 1), we establish its convergence and demonstrate that the multistage strategy are faster at initial stages. Specifically, we allow larger stepsizes in the first few stages to boost initial performance, and smaller stepsizes in the final stages decrease the size of stationary distribution. Theoretically, we properly redefine the aforementioned auxiliary sequence and Lyapunov function to incorporate the stagewise parameters.

   To the best of our knowledge, this is the first convergence guarantee for SGDM in the multistage setting.

| Method | Additional Assumptions | Convergence Bound |
|---|---|---|
| SGDM [25] | Bounded gradient | $\mathbb{E}[\|\nabla f(x_{\text{out}})\|^2] = \mathcal{O}\left(\frac{1}{k\alpha} + \frac{\alpha\sigma^2}{1-\beta}\right)$ |
| SGDM [26] | - | $\mathbb{E}[\|\nabla f(x_{\text{out}})\|^2] = \mathcal{O}\left(\frac{1}{k\alpha} + \frac{\alpha\sigma^2}{1-\beta}\right)$ |
| SGDM (*) | - | $\mathbb{E}[\|\nabla f(x_{\text{out}})\|^2] = \mathcal{O}\left(\frac{1}{k\alpha} + \alpha\sigma^2\right)$ |
| SGDM (*) | Strong convexity | $\mathbb{E}[f(x^k) - f^*] = \mathcal{O}\left((1-\alpha\mu)^k + \alpha\sigma^2\right)$ |
| Multistage SGDM(*) | - | $\mathbb{E}[\|\nabla f(x_{\text{out}})\|^2] = \mathcal{O}\left(\frac{1}{nA_2} + \frac{1}{n}\sum_{l=1}^n \alpha_l\sigma^2\right)$ |

Table 1: Comparison of our results (*) with prior work under Assumption 1 and additional assumptions. "Bounded gradient" stands for the bounded gradient assumption $\|\nabla f(x)\| \leq G$ for some $G > 0$ and all $x \in \mathbb{R}^d$. This work removes this assumption and improves convergence bounds. Strongly convex setting and multistage setting are also analyzed. We omit the results of [8] and [10] as their analysis only applies to SGD (momentum-free case).

## 1.2 Other related work

Nesterov's momentum achieves optimal convergence rate in deterministic optimization [18], and has also been combined with SGD for neural network training [24]. Recently, its multistage version has been analyzed for convex or strongly convex objectives [3, 14]. Other forms of momentum for stochastic optimization include PID Control-based methods [2], Accelerated SGD [12], and Quasi-Hyperbolic Momentum [17]. In this work, we restrict ourselves to heavy-ball momentum, which is arguably the most popular form of momentum in current deep learning practice.

## 2 Notation and Preliminaries

Throughout this paper, we use $\|\cdot\|$ for vector $\ell_2$-norm, $\langle\cdot,\cdot\rangle$ stands for dot product. Let $g^k$ denote the full gradient of $f$ at $x^k$, i.e., $g^k := \nabla f(x^k)$, and $f^* := \min_{x\in\mathbb{R}^d} f(x)$.

**Definition 1.** *We say that $f : \mathbb{R}^d \to \mathbb{R}$ is $L-$smooth with $L \geq 0$, if it is differentiable and satisfies*

$$f(y) \leq f(x) + \langle\nabla f(x), y - x\rangle + \frac{L}{2}\|y - x\|^2, \forall x, y \in \mathbb{R}^d.$$

We say that $f : \mathbb{R}^d \rightarrow \mathbb{R}$ is $\mu-$strongly convex with $\mu \geq 0$, if it satisfies

$$f(y) \geq f(x) + \langle \nabla f(x), y - x \rangle + \frac{\mu}{2} \|y - x\|^2, \forall x, y \in \mathbb{R}^d .$$

The following assumption is effective throughout, which is standard in stochastic optimization.

**Assumption 1.** *1. **Smoothness:** The objective $f(x)$ in (1) is $L-$smooth.*

    *2. **Unbiasedness:** At each iteration $k$, $\tilde{g}^k$ satisfies $\mathbb{E}_{\zeta^k}[\tilde{g}^k] = g^k$.*

    *3. **Independent samples:** the random samples $\{\zeta_k\}_{k=1}^\infty$ are independent.*

    *4. **Bounded variance:** the variance of $\tilde{g}^k$ with respect to $\zeta^k$ satisfies $\text{Var}_{\zeta^k}(\tilde{g}^k) = \mathbb{E}_{\zeta^k}[\|\tilde{g}^k - g^k\|^2] \leq \sigma^2$ for some $\sigma^2 > 0$.*

Unless otherwise noted, all the proof in the paper are deferred to the appendix.

## 3 Key Ingredients of Convergence Theory

In this section, we present some key insights for the analysis of stochastic momentum methods. For simplicity, we first focus on the case of fixed stepsize and momentum weight, and make proper generalizations for Multistage SGDM in App. C.

### 3.1 A key observation on momentum

In this section, we make the following observation on the role of momentum:

*With a momentum weight $\beta \in [0, 1)$, the update vector $m^k$ enjoys a reduced "variance" of $(1 - \beta)\sigma^2$, while having a controllable deviation from the full gradient $g^k$ in expectation.*

First, without loss of generality, we can take $m^0 = 0$, and express $m^k$ as

$$m^k = (1 - \beta) \sum_{i=1}^{k} \beta^{k-i} \tilde{g}^i. \tag{3}$$

$m^k$ is a moving average of the past stochastic gradients, with smaller weights for older ones[1].

we have the following result regarding the "variance" of $m^k$, which is measured between $m^k$ and its deterministic version $(1 - \beta) \sum_{i=1}^{k} \beta^{k-i} g^i$.

**Lemma 1.** *Under Assumption 1, the update vector $m^k$ in SGDM (2) satisfies*

$$\mathbb{E}\left[ \left\| m^k - (1 - \beta) \sum_{i=1}^{k} \beta^{k-i} g^i \right\|^2 \right] \leq \frac{1 - \beta}{1 + \beta} (1 - \beta^{2k})\sigma^2.$$

Lemma 1 follows directly from the property of the moving average.

On the other hand, $(1 - \beta) \sum_{i=1}^{k} \beta^{k-i} g^i$ is a moving average of all past gradients, which is in contrast to SGD. It seems unclear how far is $(1 - \beta) \sum_{i=1}^{k} \beta^{k-i} g^i$ from the ideal descent direction $g^k$, which could be unbounded unless stronger assumptions are imposed. Previous analysis such as [25] and [9] make the blanket assumption of bounded $\nabla f$ to circumvent this difficulty.

In this work, we provide a different perspective to resolve this issue.

**Lemma 2.** *Under Assumption 1, we have*

$$\mathbb{E}\left[ \left\| \frac{1}{1 - \beta^k} (1 - \beta) \sum_{i=1}^{k} \beta^{k-i} g^i - g^k \right\|^2 \right] \leq \sum_{i=1}^{k-1} a_{k,i} \, \mathbb{E}[\|x^{i+1} - x^i\|^2],$$

*where*

$$a_{k,i} = \frac{L^2 \beta^{k-i}}{1 - \beta^k} \left( k - i + \frac{\beta}{1 - \beta} \right). \tag{4}$$

From Lemma 2, we know the deviation of $\frac{1}{1-\beta^k}(1-\beta)\sum_{i=1}^k \beta^{k-i}g^i$ from $g^k$ is controllable sum of past successive iterate differences, in the sense that the coefficients $a_{k,i}$ decays linearly for older ones. This inspires the construction of a new Lyapunov function to handle the deviation brought by the momentum, as we shall see next.

### 3.2 A new Lyapunov function

Let us construct the following Lyapunov function for SGDM:

$$L^k = \left(f(z^k) - f^\star\right) + \sum_{i=1}^{k-1} c_i \|x^{k+1-i} - x^{k-i}\|^2. \tag{5}$$

In the Lyapunov function (5), $\{c_i\}_{i=1}^\infty$ are positive constants to be specified later corresponding to the deviation described in Lemma 2. Since the coefficients in (4) converges linearly to 0 as $k \to \infty$, we can choose $\{c_i\}_{i=1}^\infty$ in a diminishing fashion, such that this deviation can be controlled, and $L^k$ defined in (5) is indeed a Lyapunov function under strongly convex and nonconvex settings (see Propositions 1 and 2).

In (5), $z^k$ is an auxiliary sequence defined as

$$z^k = \begin{cases} x^k & k = 1, \\ \frac{1}{1-\beta}x^k - \frac{\beta}{1-\beta}x^{k-1} & k \geq 2. \end{cases} \tag{6}$$

This auxiliary sequence first appeared in the analysis of deterministic heavy ball methods in [7], and later applied in the analysis of SGDM [26, 25]. It enjoys the following property.

**Lemma 3.** $z^k$ defined in (6) satisfies

$$z^{k+1} - z^k = -\alpha \tilde{g}^k.$$

Lemma 3 indicates that it is more convenient to analyze $z^k$ than $x^k$ since $z^k$ behaves more like a SGD iterate, although the stochastic gradient $\tilde{g}^k$ is not taken at $z^k$.

Since the coefficients of the deviation in Lemma 2 converges linearly to 0 as $k \to \infty$, we can choose $\{c_i\}_{i=1}^\infty$ in a diminishing fashion, such that this deviation can be controlled. Remarkably, we shall see in Sec. 4 that with $c_1 = \mathcal{O}\left(\frac{L}{1-\beta}\right)$, $L^k$ defined in (5) is indeed a Lyapunov function under strongly convex and nonconvex settings, and that SGDM converges as fast as SGD.

Now, let us turn to the Multistage SGDM (Algorithm 1), which has been very successful in neural network training. However, its convergence still remains unclear except for the momentum-free case. To establish convergence, we require the parameters of Multistage SGDM to satisfy

$$\begin{aligned} \frac{\alpha_i \beta_i}{1-\beta_i} &\equiv A_1, \text{ for } i = 1,2,...n. \\ \alpha_i T_i &\equiv A_2, \text{ for } i = 1,2,...n. \\ 0 &\leq \beta_1 \leq \beta_2 \leq ... \leq \beta_n < 1. \end{aligned} \tag{7}$$

where $\alpha_i, \beta_i$, and $T_i$ are the stepsize, momentum weight, and stage length of $i$th stage, respectively, and $A_1, A_2$ are properly chosen constants. In principle, one applies larger stepsizes $\alpha_i$ at the initial stages, which will accelerate initial convergence, and smaller stepsizes for the final stages, which will shrink the size of final stationary distribution. As a result, (7) stipulates that less iterations are required for stages with large stepsizes and more iterations for stages with small stepsizes. Finally, (7) requires the momentum weights to be monotonically increasing, which is consistent with what's done in practice [24]. often, using constant momentum weight also works.

Under the parameter choices in (7), let us define the auxiliary sequence $z^k$ by

$$z^k = x^k - A_1 m^{k-1}. \tag{8}$$

This $\{z^k\}_{k=1}^\infty$ sequence reduces to (6) when a constant stepsize and momentum weight are applied. Furthermore, the observations made in Lemmas 1, 2, and 3 can also be generalized (see Lemmas 4, 5, 6, and 7 in App. C). In Sec. 5. we shall see that with (7) and appropriately chosen $\{c_i\}_{i=1}^\infty$, $L^k$ in (5) also defines a Lyapunov function in the multistage setting, which in turn leads to the convergence of Multistage SGDM.

## 4 Convergence of SGDM

In this section, we proceed to establish the convergence of SGDM described in (2). First, by following the idea presented in Sec. 3, we can show that $L^k$ defined in (5) is a Lyapunov function.

**Proposition 1.** *Let Assumption 1 hold. In* (2), *let* $\alpha \leq \frac{1-\beta}{2\sqrt{2}L\sqrt{\beta+\beta^2}}$. *Let* $\{c_i\}_{i=1}^{\infty}$ *in* (5) *be defined by*

$$c_1 = \frac{\frac{\beta+\beta^2}{(1-\beta)^3}L^3\alpha^2}{1 - 4\alpha^2\frac{\beta+\beta^2}{(1-\beta)^2}L^2}, \qquad c_{i+1} = c_i - \left(4c_1\alpha^2 + \frac{L\alpha^2}{1-\beta}\right)\beta^i(i + \frac{\beta}{1-\beta})L^2 \quad \text{for all } i \geq 1.$$

*Then,* $c_i > 0$ *for all* $i \geq 1$, *and*

$$\mathbb{E}[L^{k+1} - L^k] \leq \left(-\alpha + \frac{3-\beta+\beta^2}{2(1-\beta)}L\alpha^2 + 4c_1\alpha^2\right)\mathbb{E}[\|g^k\|^2] \tag{9}$$
$$+ \left(\frac{\beta^2}{2(1+\beta)}L\alpha^2\sigma^2 + \frac{1}{2}L\alpha^2\sigma^2 + 2c_1\frac{1-\beta}{1+\beta}\alpha^2\sigma^2\right).$$

By telescoping (9), we obtain the stationary convergence of SGDM under nonconvex settings.

**Theorem 1.** *Let Assumption 1 hold. In* (2), *let* $\underline{\alpha} \leq \alpha \leq \min\{\frac{1-\beta}{L(4-\beta+\beta^2)}, \frac{1-\beta}{2\sqrt{2}L\sqrt{\beta+\beta^2}}\}$. *Then,*

$$\frac{1}{k}\sum_{i=1}^{k}\mathbb{E}[\|g^i\|^2] \leq \frac{2\left(f(x^1) - f^*\right)}{k\alpha} + \left(\frac{\beta+3\beta^2}{2(1+\beta)} + 1\right)L\alpha\sigma^2 = \mathcal{O}\left(\frac{f(x^1) - f^*}{k\alpha} + L\alpha\sigma^2\right). \tag{10}$$

Now let us turn to the strongly convex setting, for which we have

**Proposition 2.** *Let Assumption 1 hold. Assume in addition that* $f$ *is* $\mu-$*strongly convex. In* (2), *let* $\alpha \leq \min\{\frac{1-\beta}{5L}, \frac{1-\beta}{L\left(3-\beta+2\beta^2+\frac{48\sqrt{\beta}}{25}\frac{2L+18\mu}{L}\right)}\}$. *Then, there exists positive constants* $c_i$ *for* (5) *such that for all* $k \geq k_0 := \lfloor\frac{\log 0.5}{\log \beta}\rfloor$, *we have*

$$\mathbb{E}[L^{k+1} - L^k] \leq -\frac{\alpha\mu}{1 + \frac{8\mu}{L}}\mathbb{E}[L^k] + (\frac{1+\beta+\beta^2}{2(1+\beta)}L + \frac{1-\beta}{1+\beta}2c_1)\alpha^2\sigma^2 + \frac{\beta^2 + \frac{L\alpha}{2}\frac{\beta^2}{1-\beta}}{(1+\frac{8\mu}{L})(1+\beta)}2\mu\alpha^2\sigma^2.$$

The choices of $\{c_i\}_{i=1}^{\infty}$ is similar to those of Proposition 1 and can be found in App. B.4. With Proposition 2, we immediately have

**Theorem 2.** *Let Assumption 1 hold and assume in addition that* $f$ *is* $\mu-$*strongly convex. Under the same settings as in Proposition 2, for all* $k \geq k_0 = \lfloor\frac{\log 0.5}{\log \beta}\rfloor$ *we have*

$$\mathbb{E}[f(z^k) - f^*] \leq \left(1 - \frac{\alpha\mu}{1 + \frac{8\mu}{L}}\right)^{k-k_0}\mathbb{E}[L^{k_0}] + \left(1 + \frac{8\mu}{L}\right)\frac{1+\beta+\beta^2}{2(1+\beta)}\frac{L}{\mu}\alpha\sigma^2$$
$$+ \left(1 + \frac{8\mu}{L}\right)\left(\frac{1}{1+\beta}\frac{12\sqrt{\beta}}{25}\frac{2L+18\mu}{\mu}\alpha\sigma^2 + \frac{\beta^2 + \frac{L\alpha}{10}\beta^2}{1 + \frac{8\mu}{L}}\frac{2}{1+\beta}\alpha\sigma^2\right)$$
$$= \mathcal{O}\left((1 - \alpha\mu)^k + \frac{L}{\mu}\alpha\sigma^2\right).$$

**Corollary 1.** *Let Assumption 1 hold and assume in addition that* $f$ *is* $\mu-$*strongly convex. Under the same settings as in Proposition 2, for all* $k \geq k_0 = \lfloor\frac{\log 0.5}{\log \beta}\rfloor$ *we have*

$$\mathbb{E}[f(x^k) - f^*] = \mathcal{O}\left(r^k + \frac{L}{\mu}\alpha\sigma^2\right),$$

*where* $r = \max\{1 - \alpha\mu, \beta\}$.

**Remark 1.**     *1. Under nonconvex settings, the classical convergence bound of SGD is $\mathcal{O}\left(\frac{f(x^1)-f^*}{k\alpha} + L\alpha\sigma^2\right)$ with $\alpha = \mathcal{O}(\frac{1}{L})$ (see, e.g., Theorem 4.8 of [4]). Therefore, Theorem 1 tells us that with $\alpha = \mathcal{O}(\frac{1-\beta}{L})$, SGDM achieves the same convergence bound as SGD.*

2. *In contrast, the radius of the stationary distribution for SGDM in [26] and [25] is $\mathcal{O}(\frac{\alpha\sigma^2}{1-\beta})$, and the latter one also assumes that $\nabla f$ is uniformly bounded.*

3. *In Theorem 2 and Corollary 1, the convergence bounds hold for $k \geq k_0 = \lfloor\frac{\log 0.5}{\log \beta}\rfloor$, where $k_0$ is a mild constant[1]. when $r = 1 - \alpha\mu$, the $\mathcal{O}(r^k)$ part in Corollary 1 matches the lower bound established in Proposition 3 of [12].*

4. *The convergence bound of SGD under strong convexity is $\mathcal{O}\left((1-\alpha\mu)^k + \frac{L}{\mu}\alpha\sigma^2\right)$ (see, e.g, Theorem 4.6 of [4]), our result for SGDM in Corollary 1 recovers this when $\beta = 0$.*

## 5    Convergence of Multistage SGDM

In this section, we switch to the Multistage SGDM (Algorithm 1).

Let us first show that when the (7) is applied, we can define the constants $c_i$ properly so that (5) still produces a Lyapunov function.

**Proposition 3.** *Let Assumption 1 hold. In Algorithm 1, let the parameters satisfy (7) with $A_1 = \frac{1}{24\sqrt{2}L}$. In addition, let*

$$\frac{1-\beta_1}{\beta_1} \leq 12\frac{1-\beta_n}{\sqrt{\beta_n+\beta_n^2}}, \qquad c_1 = \frac{\frac{\alpha_1^2}{1-\beta_1}\frac{\beta_n+\beta_n^2}{(1-\beta_n)^2}L^3}{1-4\alpha_1^2\frac{\beta_n+\beta_n^2}{(1-\beta_n)^2}L^2},$$

*and for any $i \geq 1$, let*

$$c_{i+1} = c_i - \left(4c_1\alpha_1^2 + L\frac{\alpha_1^2}{1-\beta_1}\right)\beta_n^i(i + \frac{\beta_n}{1-\beta_n})L^2.$$

*Then, we have $c_i > 0$ for any $i \geq 1$. Furthermore, with $z^k$ defined in (8), for any $k \geq 1$, we have*

$$\mathbb{E}[L^{k+1} - L^k]$$
$$\leq \left(-\alpha(k) + \frac{3 - \beta(k) + 2\beta^2(k)}{2(1-\beta(k))}L\alpha^2(k) + 4c_1\alpha^2(k)\right)\mathbb{E}[\|g^k\|^2]$$
$$+ \left(\beta^2(k)L\alpha^2(k)12\frac{\beta_1}{\sqrt{\beta_n+\beta_n^2}}\sigma^2 + \frac{1}{2}L\alpha^2(k)\sigma^2 + 4c_1(1-\beta_1)\alpha^2(k)\sigma^2\right).$$

*where $\alpha(k), \beta(k)$ are the stepsize and momentum weight applied at $k$th iteration, respectively.*

**Theorem 3.** *Let Assumption 1 hold. Under the same settings as in Proposition 3, let $\beta_1 \geq \frac{1}{2}$ and let $A_2$ be large enough such that*

$$\beta_i^{2T_i} \leq \frac{1}{2} \quad for \quad i = 1, 2, ...n.$$

*Then, we have*

$$\frac{1}{n}\sum_{l=1}^{n}\frac{1}{T_l}\sum_{i=T_1+..+T_{l-1}+1}^{T_1+..+T_l}\mathbb{E}[\|g^i\|^2] \leq \frac{2(f(x^1)-f^*)}{nA_2} + \frac{1}{n}\sum_{l=1}^{n}\left(24\beta_l^2\frac{\beta_1}{\sqrt{\beta_n+\beta_n^2}}L + 3L\right)\alpha_l\sigma^2$$
$$= \mathcal{O}\left(\frac{f(x^1)-f^*}{nA_2} + \frac{1}{n}\sum_{l=1}^{n}L\alpha_l\sigma^2\right).$$

$$(11)$$

**Remark 2.**     *1. On the left hand side of (11), we have the average of the averaged squared gradient norm of $n$ stages.*

2. *On the right hand side of* (11)*, the first term dominates at initial stages, we can apply large $\alpha_i$ for these stages to accelerate initial convergence, and use smaller $\alpha_i$ for later stages so that the size of stationary distribution is small. In contrast, (static) SGDM need to use a small stepsize $\alpha$ to make the size of stationary distribution with small.*

3. *It is unclear whether the iteration complexity of Multistage SGDM is better than SGDM or not. However, we do observe that Multistage SGDM is faster numerically. We leave the possible improved analysis of Multistage SGDM to future work.*

## 6 Experiments

In this section, we verify our theoretical claims by numerical experiments. For each combination of algorithm and training task, training is performed with 3 random seeds $1, 2, 3$. Unless otherwise stated, we report the average of losses of the past $m$ batches, where $m$ is the number of batches for the whole dataset. Our implementation is available at GitHub[1]. Additional implementation details can be found in App. E.

### 6.1 Logistic regression

**Setup.** The MNIST dataset consists of $n = 60000$ labeled examples of $28 \times 28$ gray-scale images of handwritten digits in $K = 10$ classes $0, 1, \ldots, 9$. For all algorithms, we use batch size $s = 64$ (and hence number batches per epoch is $m = 1874$), number of epochs $T = 50$. The regularization parameter is $\lambda = 5 \times 10^{-4}$.

**The effect of $\alpha$ in (static) SGDM.** By Theorem 2 we know that, with a fixed $\beta$, a larger $\alpha$ leads to faster loss decrease to the stationary distribution. However, the size of the stationary distribution is also larger. This is well illustrated in Figure 1. For example, $\alpha = 1.0$ and $\alpha = 0.5$ make losses decrease more rapidly than $\alpha = 0.1$. During later iterations, $\alpha = 0.1$ leads to a lower final loss.

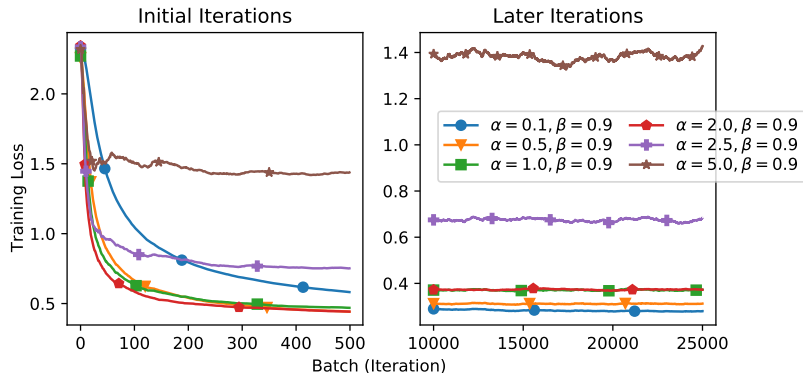

Figure 1: Logistic Regression on the MNIST Dataset using SGDM with fixed $(\alpha, \beta)$

**Multistage SGDM.** We take 3 stages for Multistage SGDM. The parameters are chosen according to (7): $T_1 = 3, T_2 = 6, T_3 = 21, \alpha_i = A_2/T_i, \beta_i = A_1/(c_2 + \alpha_i)$, where $A_2 = 2.0$ and $A_1 = 1.0$.[1] We compare Multistage SGDM with SGDM with $(\alpha, \beta) = (0.66, 0.9)$ and $(\alpha, \beta) = (0.095, 0.9)$, where $0.66, 0.095$ are the stepsizes of the first and last stage of Multistage SGDM, respectively. The training losses of initial and later iterations are shown in Figure 2.

We can see that SGDM with $(\alpha, \beta) = (0.66, 0.9)$ converges faster initially, but has a higher final loss; while SGDM with $(\alpha, \beta) = (0.095, 0.9)$ behaves the other way. Multistage SGDM takes the advantage of both, as predicted by Theorem 3. The performances of SGDM and Vanilla SGD with the same stepsize are similar.

### 6.2 Image classification

For the task of training ResNet-18 on the CIFAR-10 dataset, we compare Multistage SGDM, a baseline SGDM, and YellowFin [28], an automatic momentum tuner based on heuristics from

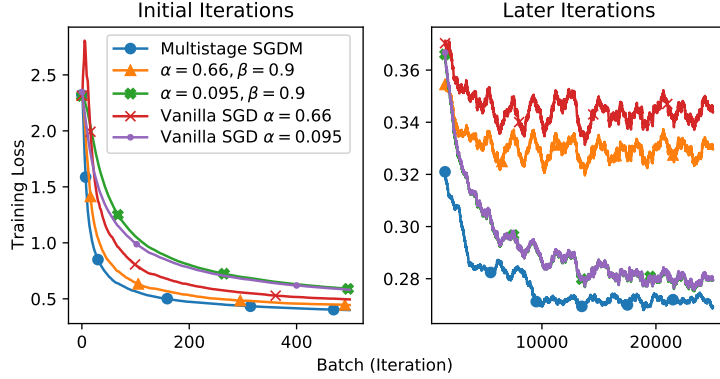

Figure 2: Logistic Regression on the MNIST using Multistage SGDM and SGDM with fixed $\beta$

optimizing strongly convex quadratics. The initial learning rate of YellowFin is set to $0.1$,[1] and other parameters are set as their default values. All algorithms are run for $T = 50$ epochs and the batch size is fixed as $s = 128$.

For Multistage SGDM, the parameters choices are governed by (7): the stage lengths are $T_1 = 5$, $T_2 = 10$, and $T_3 = 35$. Take $A_1 = 1.0$, $A_2 = 2.0$, set the per-stage stepsizes and momentum weights as $\alpha_i = A_2/T_i$ and $\beta_i = A_1/(A_1 + \alpha_i)$, for stages $i = 1, 2, 3$. For the baseline SGDM, the stepsize schedule of Multistage SGDM is applied, but with a fixed momentum $\beta = 0.9$.

In Figure 3, we present training losses and end-of-epoch validation accuracy of the tested algorithms. We can see that Multistage SGDM performs the best. Baseline SGDM is slightly worse, possibly because of its fixed momentum weight. Finally, Multistage SGDM can reach a test accuracy of $93\%$ around 200 epochs.

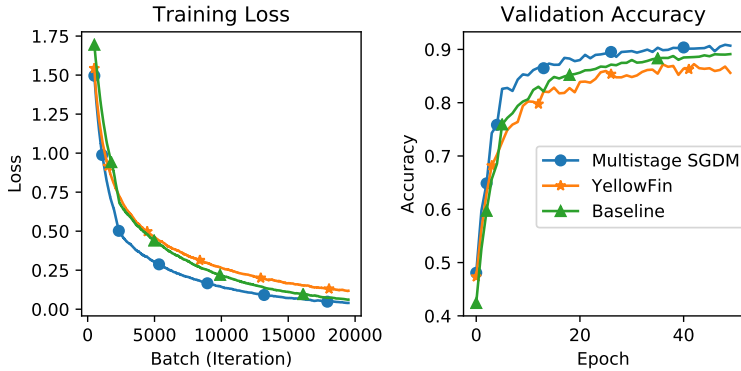

Figure 3: Training ResNet-18 on CIFAR-10

## 7 Summary and Future Directions

In this work, we provide new theoretical insights into the convergence behavior of SGDM and Multistage SGDM. For SGDM, we show that it is as fast as plain SGD in both nonconvex and strongly convex settings. For the widely adopted multistage SGDM, we establish its convergence and show the advantage of stagewise training.

There are still open problems to be addressed. For example, (a) Is it possible to show that SGDM converges faster than SGD for special objectives such as quadratic ones? (b) Are there more efficient parameter choices than (7) that guarantee even faster convergence?

## Broader Impact

The results of this paper improves the performance of stochastic gradient descent with momentum as well as its multistage version. Our study will also benefit the machine learning community. We do not believe that the results in this work will cause any ethical issue, or put anyone at a disadvantage in our society.

## Acknowledgements

Yanli Liu and Wotao Yin were partially supported by ONR Grant N000141712162. Yanli Liu was also supported by UCLA Dissertation Year Fellowship. Yuan Gao was supported by Columbia University School of Engineering and Applied Science.

## Footnotes

[1]Their implementation of SGDM does not have the $(1-\beta)$ before $\tilde{g}^k$, which gives $m^k = \sum_{i=1}^k \beta^{k-i}\tilde{g}^i$, while $m^k = (1-\beta)\sum_{i=1}^k \beta^{k-i}\tilde{g}^i$ for (2). Therefore, they only differ by a constant scaling.

[3] Here $k$ is the number of iterations. Note that in [26], a different but equivalent formulation of SGDM is analyzed; their stepsize $\gamma$ is effectively $\frac{\alpha}{1-\beta}$ in our setting.

[1]Note the sum of weights $(1 - \beta) \sum_{i=1}^{k} \beta^{k-i} = 1 - \beta^k \rightarrow 1$ as $k \rightarrow \infty$.

[1]For example, we have $k_0 = 6$ for the popular choice $\beta = 0.9$.

[1] https://github.com/gao-yuan-hangzhou/improved-analysis-sgdm

[1] Here, $A_1$ is not set by its theoretical value $\frac{1}{24L}$, since the dataset is very large and the gradient Lipschitz constant $L$ cannot be computed easily.

[1] We have experimented with initial learning rates $0.001$ (default), $0.01$, $0.1$ and $0.5$, each repeated 3 times; we found that the choice $0.1$ is the best in terms of the final training loss.

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
