[Supplementary Material]

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

Moreover, since $\zeta^1, \zeta^2, ..., \zeta^k$ are independent random variables (item 3 of Assumption 1), we can write the total expectation as $\mathbb{E} = \mathbb{E}_{\zeta^1}\mathbb{E}_{\zeta^2}...\mathbb{E}_{\zeta^k}$, and therefore

$$\mathbb{E}\left[\left\|m^k - (1-\beta)\sum_{i=1}^{k}\beta^{k-i}g^i\right\|^2\right]$$
$$= (1-\beta)^2 \mathbb{E}_{\zeta^1}\mathbb{E}_{\zeta^2}...\mathbb{E}_{\zeta^k}\left[\left\|\sum_{i=1}^{k}\beta^{k-i}(\tilde{g}^i - g^i)\right\|^2\right]$$
$$= (1-\beta)^2 \mathbb{E}_{\zeta^1}\mathbb{E}_{\zeta^2}...\mathbb{E}_{\zeta^k}\left[\sum_{i=1}^{k}\sum_{j=1}^{k}\langle \beta^{k-i}(\tilde{g}^i - g^i), \beta^{k-j}(\tilde{g}^j - g^j)\rangle\right].$$

By applying $\mathbb{E}_{\zeta^i}[\tilde{g}^i] = g^i$ (item 2 in Assumption 1), we further have for any $i > j$ that

$$\mathbb{E}_{\zeta^1}\mathbb{E}_{\zeta^2}...\mathbb{E}_{\zeta^k}\left[\langle\tilde{g}^i - \mathbb{E}_{\zeta^i}[\tilde{g}^i], \tilde{g}^j - \mathbb{E}_{\zeta^j}[\tilde{g}^j]\rangle\right]$$
$$= \mathbb{E}_{\zeta^1}\mathbb{E}_{\zeta^2}...\mathbb{E}_{\zeta^i}\left[\langle\tilde{g}^i - \mathbb{E}_{\zeta^i}[\tilde{g}^i], \tilde{g}^j - \mathbb{E}_{\zeta^j}[\tilde{g}^j]\rangle\right]$$
$$= \mathbb{E}_{\zeta^1}\mathbb{E}_{\zeta^2}...\mathbb{E}_{\zeta^{i-1}}\left[\langle\mathbb{E}_{\zeta^i}[\tilde{g}^i] - \mathbb{E}_{\zeta^i}[\tilde{g}^i], \tilde{g}^j - \mathbb{E}_{\zeta^j}[\tilde{g}^j]\rangle\right]$$
$$= 0.$$

.

It is straightforward to see that the same conclusion holds for $i < j$.

Finally, we know from the item 4 in Assumption 1 that

$$\mathbb{E}\left[\left\|m^k - (1-\beta)\sum_{i=1}^{k}\beta^{k-i}g^i\right\|^2\right]$$
$$= (1-\beta)^2 \mathbb{E}_{\zeta^1}\mathbb{E}_{\zeta^2}...\mathbb{E}_{\zeta^k}\left[\sum_{i=1}^{k}\beta^{2(k-i)}\|\tilde{g}^i - \mathbb{E}_{\zeta^i}[\tilde{g}^i]\|^2\right]$$
$$\leq \frac{1-\beta}{1+\beta}(1-\beta^{2k})\sigma^2.$$

## A.2 Proof of Lemma 2

We have

$$
\mathbb{E}\left[\left\|\frac{1-\beta}{1-\beta^k}\sum_{i=1}^{k}\beta^{k-i}g^i - g^k\right\|^2\right]
$$

$$
= \left(\frac{1-\beta}{1-\beta^k}\right)^2 \sum_{i,j=1}^{k}\mathbb{E}[\langle\beta^{k-i}(g^k-g^i),\beta^{k-j}(g^k-g^j)\rangle]
$$

$$
\leq \left(\frac{1-\beta}{1-\beta^k}\right)^2 \sum_{i,j=1}^{k}\beta^{2k-i-j}(\frac{1}{2}\mathbb{E}[\|g^k-g^i\|^2] + \frac{1}{2}\mathbb{E}[\|g^k-g^j\|^2])
$$

$$
= \left(\frac{1-\beta}{1-\beta^k}\right)^2 \sum_{i=1}^{k}\left(\sum_{j=1}^{k}\beta^{2k-i-j}\right)\frac{1}{2}\mathbb{E}[\|g^k-g^j\|^2]
$$

$$
+ \left(\frac{1-\beta}{1-\beta^k}\right)^2 \sum_{j=1}^{k}\left(\sum_{i=1}^{k}\beta^{2k-i-j}\right)\frac{1}{2}\mathbb{E}[\|g^k-g^i\|^2]
$$

$$
= \left(\frac{1-\beta}{1-\beta^k}\right)^2 \sum_{i=1}^{k}\frac{\beta^{k-i}(1-\beta^k)}{1-\beta}\mathbb{E}[\|g^k-g^i\|^2]
$$

$$
= \frac{1-\beta}{1-\beta^k}\sum_{i=1}^{k}\beta^{k-i}\mathbb{E}[\|g^k-g^i\|^2],
$$

where we have applied Cauchy-Schwarz in the first inequality.

By applying triangle inequality and the smoothness of $f$ (item 1 in Assumption 1), we further have

$$
\mathbb{E}\left[\left\|\frac{1-\beta}{1-\beta^k}\sum_{i=1}^{k}\beta^{k-i}g^i - g^k\right\|^2\right]
$$

$$
\leq \frac{1-\beta}{1-\beta^k}\sum_{i=1}^{k}\beta^{k-i}(k-i)\sum_{j=i}^{k-1}\mathbb{E}[\|g^{j+1}-g^j\|^2]
$$

$$
\leq \frac{1-\beta}{1-\beta^k}\sum_{i=1}^{k}\beta^{k-i}(k-i)\sum_{j=i}^{k-1}L^2\,\mathbb{E}[\|x^{j+1}-x^j\|^2]
$$

$$
= \frac{1-\beta}{1-\beta^k}\sum_{j=1}^{k-1}\left(\sum_{i=1}^{j}\beta^{k-i}(k-i)\right)L^2\,\mathbb{E}[\|x^{j+1}-x^j\|^2].
$$

Therefore, by defining $a'_{k,j} = \frac{1-\beta}{1-\beta^k}L^2\sum_{i=1}^{j}\beta^{k-i}(k-i)$, we get

$$
\mathbb{E}\left[\left\|\frac{1-\beta}{1-\beta^k}\sum_{i=1}^{k}\beta^{k-i}g^i - g^k\right\|^2\right] \leq \sum_{j=1}^{k-1}a'_{k,j}\,\mathbb{E}[\|x^{j+1}-x^j\|^2]. \tag{12}
$$

Furthermore, $a'_{k,j}$ can be calculated as

$$
a'_{k,j} = \frac{L^2\beta^k}{1-\beta^k}\left(-(k-1)-\frac{1}{1-\beta}\right) + \frac{L^2\beta^{k-j}}{1-\beta^k}\left(k-j+\frac{\beta}{1-\beta}\right). \tag{13}
$$

Notice that

$$
a'_{k,j} < a_{k,j} := \frac{L^2\beta^{k-j}}{1-\beta^k}\left(k-j+\frac{\beta}{1-\beta}\right). \tag{14}
$$

Combining this with (12), we finally arrive at

$$\mathbb{E}\left[\left\|\frac{1-\beta}{1-\beta^k}\sum_{i=1}^{k}\beta^{k-i}g^i - g^k\right\|^2\right] \le \sum_{i=1}^{k-1}a_{k,i}\,\mathbb{E}[\|x^{i+1}-x^i\|^2],$$

where

$$a_{k,i} = \frac{L^2\beta^{k-i}}{1-\beta^k}\left(k-i+\frac{\beta}{1-\beta}\right).$$

## A.3  Proof of Lemma 3

Let us consider the cases of $k=1$ and $k\ge 2$ separately.

For $k=1$, we have

$$z^2 - z^1 = \frac{1}{1-\beta}x^2 - \frac{\beta}{1-\beta}x^1 - x^1 = \frac{1}{1-\beta}(x^2-x^1) = -\alpha\tilde{g}^1.$$

And for $k\ge 2$, we have

$$\begin{aligned}
z^{k+1} - z^k &= \frac{1}{1-\beta}(x^{k+1}-x^k) - \frac{\beta}{1-\beta}(x^k - x^{k-1})\\
&= \frac{1}{1-\beta}(-\alpha m^k) - \frac{\beta}{1-\beta}(-\alpha m^{k-1})\\
&= \frac{1}{1-\beta}(-\alpha m^k + \alpha\beta m^{k-1})\\
&= -\alpha\tilde{g}^k.
\end{aligned}$$

# B  Main Theory for SGDM

## B.1  Objective descent

In order to prove Proposition 1, let us first show an auxiliary result.

**Proposition 4.** *Take Assumption 1. Then, for $z^k$ defined in* (6), *we have*

$$\mathbb{E}[f(z^{k+1})] \le \mathbb{E}[f(z^k)] + \left(-\alpha + \frac{1+\beta^2}{1-\beta}L\alpha^2 + \frac{1}{2}L\alpha^2\right)\mathbb{E}[\|g^k\|^2]$$

$$+ \left(\frac{\beta^2}{2(1+\beta)} + \frac{1}{2}\right)L\alpha^2\sigma^2 + \frac{\beta^2(1-\beta^k)^2 L\alpha^2}{1-\beta}\mathbb{E}\left[\left\|\frac{1-\beta}{1-\beta^k}\sum_{i=1}^{k}\beta^{k-i}g^i - g^k\right\|^2\right].$$

$$(15)$$

The smoothness of $f$ yields

$$\mathbb{E}_{\zeta^k}[f(z^{k+1})] \le f(z^k) + \mathbb{E}_{\zeta^k}[\langle\nabla f(z^k), z^{k+1}-z^k\rangle] + \frac{L}{2}\mathbb{E}_{\zeta^k}[\|z^{k+1}-z^k\|^2]$$

$$= f(z^k) + \mathbb{E}_{\zeta^k}[\langle\nabla f(z^k), -\alpha\tilde{g}^k\rangle] + \frac{L\alpha^2}{2}\mathbb{E}_{\zeta^k}[\|\tilde{g}^k\|^2],$$

$$(16)$$

where we have applied Lemma 3 in the second step.

For the inner product term, we can take full expectation $\mathbb{E} = \mathbb{E}_{\zeta^1}...\mathbb{E}_{\zeta^k}$ to get

$$\mathbb{E}[\langle\nabla f(z^k), -\alpha\tilde{g}^k\rangle] = \mathbb{E}[\langle\nabla f(z^k), -\alpha g^k\rangle],$$

which follows from the fact that $z^k$ is determined by the previous $k-1$ random samples $\zeta^1, \zeta^2, ...\zeta^{k-1}$, which is independent of $\zeta^k$, and $\mathbb{E}_{\zeta^k}[\tilde{g}^k] = g^k$.

So, we can bound

$$\mathbb{E}[\langle \nabla f(z^k), -\alpha \tilde{g}^k \rangle] = \mathbb{E}[\langle \nabla f(z^k) - g^k, -\alpha g^k \rangle] - \alpha \mathbb{E}[\|g^k\|^2]$$
$$\leq \alpha \frac{\rho_0}{2} L^2 \mathbb{E}[\|z^k - x^k\|^2] + \alpha \frac{1}{2\rho_0} \mathbb{E}[\|g^k\|^2] - \alpha \mathbb{E}[\|g^k\|^2],$$

where $\rho_0 > 0$ can be any positive constant (to be determined later).

Combining (16) and the last inequality, we arrive at

$$\mathbb{E}[f(z^{k+1})] \leq \mathbb{E}[f(z^k)] + \alpha \frac{\rho_0}{2} L^2 \mathbb{E}[\|z^k - x^k\|^2]$$
$$+ (\alpha \frac{1}{2\rho_0} - \alpha) \mathbb{E}[\|g^k\|^2] + \frac{L\alpha^2}{2} \mathbb{E}[\|\tilde{g}^k\|^2].$$

Since $z^k = x^k$ when $k = 1$ and $z^k = \frac{1}{1-\beta} x^k - \frac{\beta}{1-\beta} x^{k-1}$ when $k \geq 2$, it can be verified that $z^k - x^k = -\frac{\beta}{1-\beta} \alpha m^{k-1}$. Consequently,

$$\mathbb{E}[f(z^{k+1})] \leq \mathbb{E}[f(z^k)] + \alpha^3 \frac{\rho_0}{2} L^2 (\frac{\beta}{1-\beta})^2 \mathbb{E}[\|m^{k-1}\|^2]$$
$$+ (\alpha \frac{1}{2\rho_0} - \alpha) \mathbb{E}[\|g^k\|^2] + \frac{L\alpha^2}{2} \mathbb{E}[\|\tilde{g}^k\|^2]. \tag{17}$$

On the other hand, from Lemma 1 we know that

$$\mathbb{E}[\|m^{k-1}\|^2] \leq 2 \mathbb{E}[\|m^{k-1} - (1-\beta) \sum_{i=1}^{k-1} \beta^{k-1-i} g^i\|^2] + 2 \mathbb{E}[\|(1-\beta) \sum_{i=1}^{k-1} \beta^{k-1-i} g^i\|^2]$$
$$\leq 2 \frac{1-\beta}{1+\beta} \sigma^2 + 2 \mathbb{E}[\|(1-\beta) \sum_{i=1}^{k-1} \beta^{k-1-i} g^i\|^2]$$
$$\mathbb{E}[\|\frac{1-\beta}{1-\beta^{k-1}} \sum_{i=1}^{k-1} \beta^{k-1-i} g^i\|^2] \leq 2 \mathbb{E}[\|g^k\|^2] + 2 \mathbb{E}[\|\frac{1-\beta}{1-\beta^{k-1}} \sum_{i=1}^{k-1} \beta^{k-1-i} g^i - g^k\|^2],$$
$$\mathbb{E}[\|\tilde{g}^k\|^2] \leq \sigma^2 + \mathbb{E}[\|g^k\|^2]. \tag{18}$$

Putting these into (17), we arrive at

$$\mathbb{E}[f(z^{k+1})] \leq \mathbb{E}[f(z^k)] + \left( -\alpha + \alpha \frac{1}{2\rho_0} + 2\alpha^3 \rho_0 L^2 (\frac{\beta}{1-\beta})^2 (1 - \beta^{k-1})^2 + \frac{L\alpha^2}{2} \right) \mathbb{E}[\|g^k\|^2]$$
$$+ \left( \alpha^3 \rho_0 L^2 (\frac{\beta}{1-\beta})^2 \frac{1-\beta}{1+\beta} \sigma^2 + \frac{L\alpha^2}{2} \sigma^2 \right)$$
$$+ 2\alpha^3 \rho_0 L^2 (\frac{\beta}{1-\beta})^2 (1 - \beta^{k-1})^2 \mathbb{E}[\|\frac{1-\beta}{1-\beta^{k-1}} \sum_{i=1}^{k-1} \beta^{k-1-i} g^i - g^k\|^2].$$

Substituting

$$\mathbb{E}[\|\frac{1-\beta}{1-\beta^k} \sum_{i=1}^{k} \beta^{k-i} g^i - g^k\|^2] = \mathbb{E}[\|\frac{1}{1-\beta^k} \beta(1-\beta) \sum_{i=1}^{k-1} \beta^{k-1-i} g^i - \frac{1-\beta^{k-1}}{1-\beta^k} \beta g^k\|^2]$$
$$= \beta^2 (\frac{1-\beta^{k-1}}{1-\beta^k})^2 \mathbb{E}[\|\frac{1-\beta}{1-\beta^{k-1}} \sum_{i=1}^{k-1} \beta^{k-1-i} g^i - g^k\|^2]$$

into the last inequality produces

$$\mathbb{E}[f(z^{k+1})] \le \mathbb{E}[f(z^k)] + \left( -\alpha + \alpha\frac{1}{2\rho_0} + 2\alpha^3\rho_0 L^2(\frac{\beta}{1-\beta})^2(1-\beta^{k-1})^2 + \frac{L\alpha^2}{2} \right)\mathbb{E}[\|g^k\|^2]$$
$$+ \left( \alpha^3\rho_0 L^2(\frac{\beta}{1-\beta})^2\frac{1-\beta}{1+\beta}\sigma^2 + \frac{L\alpha^2}{2}\sigma^2 \right)$$
$$+ 2\alpha^3\rho_0 L^2(\frac{1}{1-\beta})^2(1-\beta^k)^2\,\mathbb{E}[\|\frac{1-\beta}{1-\beta^k}\sum_{i=1}^k\beta^{k-i}g^i - g^k\|^2].$$

$$(19)$$

Finally, using $1-\beta^{k-1} < 1$ and $\rho_0 = \frac{1-\beta}{2L\alpha}$ gives

$$\mathbb{E}[f(z^{k+1})] \le \mathbb{E}[f(z^k)] + (-\alpha + \frac{1+\beta^2}{1-\beta}L\alpha^2 + \frac{1}{2}L\alpha^2)\,\mathbb{E}[\|g^k\|^2]$$
$$+ (\frac{\beta^2}{2(1+\beta)} + \frac{1}{2})L\alpha^2\sigma^2 + \frac{\beta^2(1-\beta^k)^2 L\alpha^2}{1-\beta}\,\mathbb{E}\left[\left\|\frac{1-\beta}{1-\beta^k}\sum_{i=1}^k\beta^{k-i}g^i - g^k\right\|^2\right].$$

### B.2  Proof of Proposition 1

Recall that $L^k$ is defined as

$$L^k = f(z^k) - f^* + \sum_{i=1}^{k-1}c_i\|x^{k+1-i} - x^{k-i}\|^2,$$

Therefore, by (19) we know that

$$\mathbb{E}[L^{k+1} - L^k] \le \left( -\alpha + \alpha\frac{1}{2\rho_0} + 2\alpha^3\rho_0 L^2(\frac{\beta}{1-\beta})^2 + \frac{L\alpha^2}{2} \right)\mathbb{E}[\|g^k\|^2]$$
$$+ \left( \alpha^3\rho_0 L^2(\frac{\beta}{1-\beta})^2\frac{1-\beta}{1+\beta}\sigma^2 + \frac{1}{2}L\alpha^2\sigma^2 \right)$$
$$+ \sum_{i=1}^{k-1}(c_{i+1} - c_i)\,\mathbb{E}[\|x^{k+1-i} - x^{k-i}\|^2] \qquad (20)$$
$$+ c_1\,\mathbb{E}[\|x^{k+1} - x^k\|^2]$$
$$+ 2\alpha^3\rho_0 L^2(\frac{1}{1-\beta})^2(1-\beta^k)^2\,\mathbb{E}[\|\frac{1-\beta}{1-\beta^k}\sum_{i=1}^k\beta^{k-i}g^i - g^k\|^2],$$

where $\rho_0 = \frac{1-\beta}{2L\alpha}$.

To bound the $c_1\,\mathbb{E}[\|x^{k+1} - x^k\|^2]$ term, we need to following inequalities, which are obtained in a similar way as (18).

$$\mathbb{E}[\|m^k\|^2] \le 2\,\mathbb{E}[\|m^k - (1-\beta)\sum_{i=1}^k\beta^{k-i}g^i\|^2] + 2\,\mathbb{E}[\|(1-\beta)\sum_{i=1}^k\beta^{k-i}g^i\|^2]$$

$$\le 2\frac{1-\beta}{1+\beta}\sigma^2 + 2\,\mathbb{E}[\|(1-\beta)\sum_{i=1}^k\beta^{k-i}g^i\|^2]$$

$$\mathbb{E}[\|\frac{1-\beta}{1-\beta^k}\sum_{i=1}^k\beta^{k-i}g^i\|^2] \le 2\,\mathbb{E}[\|g^k\|^2] + 2\,\mathbb{E}[\|\frac{1-\beta}{1-\beta^k}\sum_{i=1}^k\beta^{k-i}g^i - g^k\|^2],$$

$$\mathbb{E}[\|\tilde{g}^k\|^2] \le \sigma^2 + \mathbb{E}[\|g^k\|^2].$$

$$(21)$$

Therefore, $c_1 \mathbb{E}[\|x^{k+1} - x^k\|^2]$ can be bounded as

$$c_1 \mathbb{E}[\|x^{k+1} - x^k\|^2] = c_1 \alpha^2 \mathbb{E}[\|m^k\|^2]$$

$$\leq c_1 \alpha^2 \left( 2\frac{1-\beta}{1+\beta}\sigma^2 + 4\mathbb{E}[\|g^k\|^2](1-\beta^k)^2 \right)$$

$$+ 4c_1 \alpha^2 \mathbb{E}[\|\frac{1-\beta}{1-\beta^k}\sum_{i=1}^{k}\beta^{k-i}g^i - g^k\|^2]$$

$$< c_1 \alpha^2 \left( 2\frac{1-\beta}{1+\beta}\sigma^2 + 4\mathbb{E}[\|g^k\|^2] \right)$$

$$+ 4c_1 \alpha^2 (1-\beta^k)^2 \mathbb{E}[\|\frac{1-\beta}{1-\beta^k}\sum_{i=1}^{k}\beta^{k-i}g^i - g^k\|^2]$$

Combine this with (20), we obtain

$$\mathbb{E}[L^{k+1} - L^k]$$

$$\leq \left( -\alpha + \alpha\frac{1}{2\rho_0} + 2\alpha^3\rho_0 L^2 (\frac{\beta}{1-\beta})^2 + \frac{L\alpha^2}{2} + 4c_1\alpha^2 \right) \mathbb{E}[\|g^k\|^2]$$

$$+ \left( \alpha^3\rho_0 L^2 (\frac{\beta}{1-\beta})^2 \frac{1-\beta}{1+\beta}\sigma^2 + \frac{1}{2}L\alpha^2\sigma^2 + 2c_1\frac{1-\beta}{1+\beta}\alpha^2\sigma^2 \right)$$

$$+ \sum_{i=1}^{k-1}(c_{i+1} - c_i)\,\mathbb{E}[\|x^{k+1-i} - x^{k-i}\|^2] \tag{22}$$

$$+ 4c_1\alpha^2(1-\beta^k)^2\,\mathbb{E}[\|\frac{1-\beta}{1-\beta^k}\sum_{i=1}^{k}\beta^{k-i}g^i - g^k\|^2]$$

$$+ 2\alpha^3\rho_0 L^2(\frac{1}{1-\beta})^2(1-\beta^k)^2\,\mathbb{E}[\|\frac{1-\beta}{1-\beta^k}\sum_{i=1}^{k}\beta^{k-i}g^i - g^k\|^2].$$

In the rest of the proof, let us show that the sum of the last three terms in (22) is non-positive.
First of all, by Lemma 2 we know that

$$\mathbb{E}\left[ \left\| \frac{1}{1-\beta^k}(1-\beta)\sum_{i=1}^{k}\beta^{k-i}g^i - g^k \right\|^2 \right] \leq \sum_{i=1}^{k-1}a_{k,i}\,\mathbb{E}[\|x^{i+1} - x^i\|^2],$$

where

$$a_{k,i} = \frac{L^2\beta^{k-i}}{1-\beta^k}\left( k - i + \frac{\beta}{1-\beta} \right).$$

Or equivalently,

$$\mathbb{E}\left\| \frac{1}{1-\beta^k}(1-\beta)\sum_{i=1}^{k}\beta^{k-i}g^i - g^k \right\|^2 \leq \sum_{i=1}^{k-1}a_{k,k-i}\,\mathbb{E}\|x^{k+1-i} - x^{k-i}\|^2,$$

where

$$a_{k,k-i} = \frac{L^2\beta^i}{1-\beta^k}\left( i + \frac{\beta}{1-\beta} \right).$$

Therefore, in order to make the sum of the last three terms of (22) to be non-positive, we need to have

$$c_{i+1} \leq c_i - \left( 4c_1\alpha^2(1-\beta^k)^2 + 2\alpha^3\rho_0 L^2\frac{(1-\beta^k)^2}{(1-\beta)^2} \right)a_{k,k-i}$$

for all $i \geq 1$.

Since $1 - \beta^k < 1$, it suffices to enforce the following for all $i \geq 1$:

$$c_{i+1} = c_i - \left( 4c_1\alpha^2 + 2\alpha^3\rho_0 L^2 \frac{1}{(1-\beta)^2} \right) \beta^i (i + \frac{\beta}{1-\beta}) L^2. \tag{23}$$

And in order for $c_i > 0$ for all $i \geq 1$, we can determine $c_1$ by

$$c_1 = \left( 4c_1\alpha^2 + 2\alpha^3\rho_0 L^2 \frac{1}{(1-\beta)^2} \right) \sum_{i=1}^{\infty} \beta^i (i + \frac{\beta}{1-\beta}) L^2.$$

Since

$$\sum_{i=1}^{j} i\beta^i = \frac{1}{1-\beta} \left( \frac{\beta(1-\beta^j)}{1-\beta} - j\beta^{j+1} \right),$$

we have $\sum_{i=1}^{\infty} i\beta^i = \frac{\beta}{(1-\beta)^2}$ and

$$c_1 = \left( 4c_1\alpha^2 + 2\alpha^3\rho_0 L^2 \frac{1}{(1-\beta)^2} \right) \frac{\beta + \beta^2}{(1-\beta)^2} L^2.$$

This stipulates that

$$c_1 = \frac{2\alpha^3\rho_0 L^4 \frac{\beta+\beta^2}{(1-\beta)^4}}{1 - 4\alpha^2 \frac{\beta+\beta^2}{(1-\beta)^2} L^2}. \tag{24}$$

Notice that $\alpha \leq \frac{1-\beta}{4L\sqrt{\beta+\beta^2}}$ ensures $c_1 > 0$.

With the choices of $c_i$ in (23) and (24), the sum of the last three terms of (22) is non-positive. Therefore,

$$\mathbb{E}[L^{k+1} - L^k] \leq \left( -\alpha + \alpha\frac{1}{2\rho_0} + 2\alpha^3\rho_0 L^2(\frac{\beta}{1-\beta})^2 + \frac{L\alpha^2}{2} + 4c_1\alpha^2 \right) \mathbb{E}[\|g^k\|^2]$$
$$+ \left( \alpha^3\rho_0 L^2(\frac{\beta}{1-\beta})^2 \frac{1-\beta}{1+\beta}\sigma^2 + \frac{1}{2}L\alpha^2\sigma^2 + 2c_1\frac{1-\beta}{1+\beta}\alpha^2\sigma^2 \right). \tag{25}$$

Finally, taking

$$\rho_0 = \frac{1-\beta}{2L\alpha} \tag{26}$$

in (24), (23), and (25) gives

$$c_1 = \frac{\frac{\beta+\beta^2}{(1-\beta)^3}L^3\alpha^2}{1 - 4\alpha^2\frac{\beta+\beta^2}{(1-\beta)^2}L^2},$$

$$c_{i+1} = c_i - \left( 4c_1\alpha^2 + \frac{L\alpha^2}{(1-\beta)} \right) \beta^i (i + \frac{\beta}{1-\beta}) L^2,$$

$$\mathbb{E}[L^{k+1} - L^k] \leq \left( -\alpha + \frac{3-\beta+2\beta^2}{2(1-\beta)}L\alpha^2 + 4c_1\alpha^2 \right) \mathbb{E}[\|g^k\|^2]$$
$$+ \left( \frac{\beta^2}{2(1+\beta)}L\alpha^2\sigma^2 + \frac{1}{2}L\alpha^2\sigma^2 + 2c_1\frac{1-\beta}{1+\beta}\alpha^2\sigma^2 \right).$$

### B.3 Proof of Theorem 1

From (25) we know that

$$\mathbb{E}[L^{k+1} - L^k] \leq -R_1 \mathbb{E}[\|g^k\|^2] + R_2, \tag{27}$$

where

$$R_1 = \alpha - \alpha \frac{1}{2\rho_0} - 2\alpha^3 \rho_0 L^2 (\frac{\beta}{1-\beta})^2 - \frac{L\alpha^2}{2} - 4c_1\alpha^2 \tag{28}$$

$$R_2 = \alpha^3 \rho_0 L^2 (\frac{\beta}{1-\beta})^2 \frac{1-\beta}{1+\beta}\sigma^2 + \frac{1}{2}L\alpha^2\sigma^2 + 2c_1\frac{1-\beta}{1+\beta}\alpha^2\sigma^2, \tag{29}$$

and $\rho_0 = \frac{1-\beta}{2L\alpha}$.

This immediately tells us that

$$L^1 \geq \mathbb{E}[L^1 - L^{k+1}] \geq R_1 \sum_{i=1}^{k} \mathbb{E}[\|g^i\|^2] - kR_2,$$

and therefore

$$\frac{1}{k}\sum_{i=1}^{k}\mathbb{E}[\|g^k\|^2] \leq \frac{L^1}{kR_1} + \frac{R_2}{R_1}. \tag{30}$$

In the rest the proof, we will bound $R_1$ and $R_2$ appropriately.

First, let us show that $R_1 \geq \frac{\alpha}{2}$ when $\rho_0 = \frac{1-\beta}{2L\alpha}$ as in (26) and $\alpha \leq \min\{\frac{1-\beta}{L(4-\beta+\beta^2)}, \frac{1-\beta}{2L\sqrt{\beta+\beta^2}}\}$.

From (24) we know that

$$c_1 = \frac{2\alpha^3\rho_0 L^4 \frac{\beta+\beta^2}{(1-\beta)^4}}{1 - 4\alpha^2\frac{\beta+\beta^2}{(1-\beta)^2}L^2}.$$

Since $\alpha \leq \frac{1-\beta}{2\sqrt{2}L\sqrt{\beta+\beta^2}}$, we have

$$4\alpha^2 \frac{\beta+\beta^2}{(1-\beta)^2}L^2 \leq \frac{1}{2} \tag{31}$$

and

$$c_1 \leq 4\alpha^3 \rho_0 L^4 \frac{\beta+\beta^2}{(1-\beta)^4} \leq \frac{1}{4}\alpha\rho_0 \frac{L^2}{(1-\beta)^2}. \tag{32}$$

Therefore, in order to ensure $R_1 \geq \frac{\alpha}{2}$ where $R_1$ is defined in (28), it suffices to have

$$\alpha \frac{1}{2\rho_0} + 2\alpha\rho_0 L^2 (\frac{\beta}{1-\beta})^2\alpha^2 + \frac{L\alpha^2}{2} + \alpha\rho_0 L^2 \frac{1}{(1-\beta)^2}\alpha^2 \leq \frac{\alpha}{2}. \tag{33}$$

Applying $\rho_0 = \frac{1-\beta}{2L\alpha}$ yields

$$\alpha \frac{1}{2\rho_0} + 2\alpha\rho_0 L^2 (\frac{\beta}{1-\beta})^2\alpha^2 + \frac{L\alpha^2}{2} + \alpha\rho_0 L^2 \frac{1}{(1-\beta)^2}\alpha^2$$

$$= \frac{L\alpha^2}{1-\beta} + \alpha^2 L\frac{\beta^2}{1-\beta} + \frac{1}{2}\alpha^2 L\frac{1}{1-\beta} + \frac{L\alpha^2}{2}$$

$$= L\alpha^2 \left( \frac{1}{1-\beta} + \frac{\beta^2}{1-\beta} + \frac{1}{2}\frac{1}{1-\beta} + \frac{1}{2} \right)$$

$$= L\alpha^2 \frac{4-\beta+2\beta^2}{2(1-\beta)}$$

$$\leq \frac{\alpha}{2},$$

where we have applied $\alpha \leq \frac{1-\beta}{L(4-\beta+2\beta^2)}$ in the last step.

Therefore, (33) is true and

$$R_1 \geq \frac{\alpha}{2}. \tag{34}$$

Now let us turn to $R_2$. By (32) we know that

$$R_2 = \alpha \rho_0 L^2 (\frac{\beta}{1-\beta})^2 \alpha^2 \frac{1-\beta}{1+\beta}\sigma^2 + \frac{1}{2}L\alpha^2\sigma^2 + 2c_1\frac{1-\beta}{1+\beta}\alpha^2\sigma^2$$

$$\leq \alpha \rho_0 L^2 (\frac{\beta}{1-\beta})^2 \alpha^2 \frac{1-\beta}{1+\beta}\sigma^2 + \frac{1}{2}L\alpha^2\sigma^2 + 8\alpha^3\rho_0 L^4 \frac{\beta+\beta^2}{(1-\beta)^4}\frac{1-\beta}{1+\beta}\alpha^2\sigma^2.$$

Since $\rho_0 = \frac{1-\beta}{2L\alpha}$, we have

$$R_2 \leq \frac{\beta^2}{2(1+\beta)}L\alpha^2\sigma^2 + \frac{1}{2}L\alpha^2\sigma^2 + \frac{4\beta}{(1-\beta)^2}L^3\alpha^4\sigma^2.$$

By applying $\alpha \leq \min\{\frac{1-\beta}{L(4-\beta+\beta^2)}, \frac{1-\beta}{2\sqrt{2}L\sqrt{\beta+\beta^2}}\} \leq \frac{1-\beta}{3.75L} < \frac{1-\beta}{4L}$, we further have

$$R_2 \leq \frac{\beta^2}{2(1+\beta)}L\alpha^2\sigma^2 + \frac{1}{2}L\alpha^2\sigma^2 + \frac{\beta}{4}L\alpha^2\sigma^2. \tag{35}$$

By putting (34) and (35) into (30), we finally obtain

$$\frac{1}{k}\sum_{i=1}^{k}\mathbb{E}[\|g^i\|^2] \leq \frac{2\left(f(x^1)-f^*\right)}{k\alpha} + \left(\frac{\beta+3\beta^2}{2(1+\beta)}+1\right)L\alpha\sigma^2$$

$$= \mathcal{O}\left(\frac{f(x^1)-f^*}{k\alpha}\right) + \mathcal{O}\left(L\alpha\sigma^2\right).$$

## B.4 Proof of Proposition 2

In order to prove Proposition 2, we will set

$$c_1 = \left(\frac{\sqrt{\beta}}{(1-\sqrt{\beta})^2} + \frac{\sqrt{\beta}}{1-\sqrt{\beta}}\frac{\beta}{1-\beta}\right)\left(\frac{2L^3\alpha^2}{1-\beta} + \frac{18L^2\mu\alpha^2}{(1-\beta)(1+\frac{8\mu}{L})}\right),$$

$$c_{i+1} - c_i + A_3 2L^2\beta^{k-i}\left(k-i+\frac{\beta}{1-\beta}\right) = A_1 c_i, \qquad \forall i \geq 1,$$

Take $\rho_0 = \frac{1-\beta}{2L\alpha}$ in (22), we have

$$\mathbb{E}[L^{k+1}-L^k] \leq \left(-\alpha + \frac{3-\beta+2\beta^2}{2(1-\beta)}L\alpha^2 + 4c_1\alpha^2\right)\mathbb{E}[\|g^k\|^2]$$

$$+ \left(\frac{\beta^2}{2(1+\beta)}L\alpha^2\sigma^2 + \frac{1}{2}L\alpha^2\sigma^2 + 2c_1\frac{1-\beta}{1+\beta}\alpha^2\sigma^2\right)$$

$$+ \sum_{i=1}^{k-1}(c_{i+1}-c_i)\mathbb{E}[\|x^{k+1-i}-x^{k-i}\|^2] \tag{36}$$

$$+ \left(4c_1\alpha^2 + \frac{L\alpha^2}{(1-\beta)}\right)(1-\beta^k)^2\mathbb{E}[\|\frac{1}{1-\beta^k}(1-\beta)\sum_{i=1}^{k}\beta^{k-i}g^i - g^k\|^2].$$

Let us first derive a lower bound of the first term on the right hand side of (36).

From the strong convexity of $f$ we have

$$\mathbb{E}[\|g^k\|^2] = \mathbb{E}[\|\nabla f(x^k)\|^2] \geq 2\mu\,\mathbb{E}[f(x^k)-f^*], \tag{37}$$

where $f^\star = \min_{x \in \mathbb{R}^d} f(x)$. On the other hand, for $\mathbb{E}[f(x^k)]$ we have

$$\mathbb{E}[f(z^k)] \le \mathbb{E}[f(x^k)] + \mathbb{E}[\langle g^k, z^k - x^k \rangle] + \frac{L}{2} \mathbb{E}[\|z^k - x^k\|^2]$$

$$= \mathbb{E}[f(x^k)] + \mathbb{E}[\langle g^k - \frac{1}{1-\beta^k}(1-\beta)\sum_{i=1}^k \beta^{k-i}g^i + \frac{1}{1-\beta^k}(1-\beta)\sum_{i=1}^k \beta^{k-i}g^i, -\frac{\alpha\beta}{1-\beta}m^{k-1}\rangle]$$

$$+ \frac{L}{2} \mathbb{E}[\|\frac{\alpha\beta}{1-\beta}m^{k-1}\|^2]$$

$$\le \mathbb{E}[f(x^k)] + \alpha\frac{\rho}{2} \mathbb{E}[\|g^k - \frac{1}{1-\beta^k}(1-\beta)\sum_{i=1}^k \beta^{k-i}g^i\|^2] + \frac{\alpha}{2\rho} \mathbb{E}[\|\frac{\beta}{1-\beta}m^{k-1}\|^2]$$

$$+ \mathbb{E}[\langle \frac{1}{1-\beta^k}(1-\beta)\sum_{i=1}^k \beta^{k-i}g^i, -\frac{\alpha\beta}{1-\beta}m^{k-1}\rangle] + \frac{L}{2} \mathbb{E}[\|\frac{\alpha\beta}{1-\beta}m^{k-1}\|^2]$$

$$\le \mathbb{E}[f(x^k)] + \alpha\frac{\rho}{2} \mathbb{E}[\|g^k - \frac{1}{1-\beta^k}(1-\beta)\sum_{i=1}^k \beta^{k-i}g^i\|^2]$$

$$+ \left( \frac{\alpha}{2\rho}(\frac{\beta}{1-\beta})^2 + \frac{L\alpha^2}{2}(\frac{\beta}{1-\beta})^2 \right) \mathbb{E}[\|m^{k-1}\|^2]$$

$$+ \alpha\frac{\beta}{1-\beta} \left( \frac{\rho_1}{2} \mathbb{E}[\|\frac{1}{1-\beta^k}(1-\beta)\sum_{i=1}^k \beta^{k-i}g^i\|^2] + \frac{1}{2\rho_1} \mathbb{E}[\|m^{k-1}\|^2] \right)$$

$$= \mathbb{E}[f(x^k)] + \alpha\frac{\rho}{2} \mathbb{E}[\|g^k - \frac{1}{1-\beta^k}(1-\beta)\sum_{i=1}^k \beta^{k-i}g^i\|^2]$$

$$+ \left( \frac{\alpha}{2\rho}(\frac{\beta}{1-\beta})^2 + \frac{L\alpha^2}{2}(\frac{\beta}{1-\beta})^2 + \alpha\frac{\beta}{1-\beta}\frac{1}{2\rho_1} \right) \mathbb{E}[\|m^{k-1}\|^2]$$

$$+ \alpha\frac{\beta}{1-\beta}\frac{\rho_1}{2} \mathbb{E}[\|\frac{1}{1-\beta^k}(1-\beta)\sum_{i=1}^k \beta^{k-i}g^i\|^2],$$

where $\rho, \rho_1 > 0$ are to be determined later.

Combining this with (37) gives

$$\mathbb{E}[\|g^k\|^2] \ge 2\mu\left( \mathbb{E}[f(z^k)] - f^\star - \alpha\frac{\rho}{2} \mathbb{E}[\|g^k - \frac{1}{1-\beta^k}(1-\beta)\sum_{i=1}^k \beta^{k-i}g^i\|^2] \right.$$

$$- \left( \frac{\alpha}{2\rho}(\frac{\beta}{1-\beta})^2 + \frac{L\alpha^2}{2}(\frac{\beta}{1-\beta})^2 + \alpha\frac{\beta}{1-\beta}\frac{1}{2\rho_1} \right) \mathbb{E}[\|m^{k-1}\|^2] \qquad (38)$$

$$\left. - \alpha\frac{\beta}{1-\beta}\frac{\rho_1}{2} \mathbb{E}[\|\frac{1}{1-\beta^k}(1-\beta)\sum_{i=1}^k \beta^{k-i}g^i\|^2], \right).$$

On the other hand, we have from (18) that

$$\mathbb{E}[\|m^{k-1}\|^2] \le 2\frac{1-\beta}{1+\beta}\sigma^2 + 2(1-\beta^{k-1})^2 \left( 2\mathbb{E}[\|g^k\|^2] + 2\mathbb{E}[\|\frac{1}{1-\beta^{k-1}}(1-\beta)\sum_{i=1}^{k-1} \beta^{k-1-i}g^i - g^k\|^2] \right)$$

$$= 2\frac{1-\beta}{1+\beta}\sigma^2$$

$$+ 2(1-\beta^{k-1})^2 \left( 2\mathbb{E}[\|g^k\|^2] + 2\frac{1}{\beta^2}(\frac{1-\beta^k}{1-\beta^{k-1}})^2 \mathbb{E}[\|\frac{1}{1-\beta^k}(1-\beta)\sum_{i=1}^k \beta^{k-i}g^i - g^k\|^2] \right),$$

and that

$$\mathbb{E}[\|\frac{1}{1-\beta^k}(1-\beta)\sum_{i=1}^k \beta^{k-i}g^i\|^2] \le 2\mathbb{E}[\|g^k\|^2] + 2\mathbb{E}[\|\frac{1}{1-\beta^k}(1-\beta)\sum_{i=1}^k \beta^{k-i}g^i - g^k\|^2].$$

Putting these two inequalities into (38) and rearranging gives

$$
\left[1 + 2\mu\left(\left(\frac{\alpha}{2\rho}(\frac{\beta}{1-\beta})^2 + \frac{L\alpha^2}{2}(\frac{\beta}{1-\beta})^2 + \alpha\frac{\beta}{1-\beta}\frac{1}{2\rho_1}\right)4(1-\beta^{k-1})^2\right.\right.
$$

$$
\left.\left. + \alpha\frac{\beta}{1-\beta}\rho_1\right)\right]\mathbb{E}[\|g^k\|^2]
$$

$$
\geq 2\mu\left[\mathbb{E}[f(z^k)] - f^\star - \alpha\frac{\rho}{2}\mathbb{E}[\|g^k - \frac{1}{1-\beta^k}(1-\beta)\sum_{i=1}^{k}\beta^{k-i}g^i\|^2]\right.
$$

$$
- \left(\frac{\alpha}{2\rho}(\frac{\beta}{1-\beta})^2 + \frac{L\alpha^2}{2}(\frac{\beta}{1-\beta})^2 + \alpha\frac{\beta}{1-\beta}\frac{1}{2\rho_1}\right)
$$

$$
\times\left(2\frac{1-\beta}{1+\beta}\sigma^2 + (1-\beta^{k-1})^24\frac{1}{\beta^2}(\frac{1-\beta^k}{1-\beta^{k-1}})^2\mathbb{E}[\|\frac{1}{1-\beta^k}(1-\beta)\sum_{i=1}^{k}\beta^{k-i}g^i - g^k\|^2]\right)
$$

$$
\left. - \alpha\frac{\beta}{1-\beta}\frac{\rho_1}{2}2\mathbb{E}[\|\frac{1}{1-\beta^k}(1-\beta)\sum_{i=1}^{k}\beta^{k-i}g^i - g^k\|^2]\right]
$$

$$
= 2\mu\left[\mathbb{E}[f(z^k)] - f^\star\right.
$$

$$
- \left(\frac{\alpha}{2\rho}(\frac{\beta}{1-\beta})^2 + \frac{L\alpha^2}{2}(\frac{\beta}{1-\beta})^2 + \alpha\frac{\beta}{1-\beta}\frac{1}{2\rho_1}\right)2\frac{1-\beta}{1+\beta}\sigma^2
$$

$$
- \left(\alpha\frac{\rho}{2} + \left(\frac{\alpha}{2\rho}(\frac{\beta}{1-\beta})^2 + \frac{L\alpha^2}{2}(\frac{\beta}{1-\beta})^2 + \alpha\frac{\beta}{1-\beta}\frac{1}{2\rho_1}\right)4(1-\beta^k)^2\frac{1}{\beta^2} + \alpha\frac{\beta}{1-\beta}\rho_1\right)
$$

$$
\left.\times\mathbb{E}[\|\frac{1}{1-\beta^k}(1-\beta)\sum_{i=1}^{k}\beta^{k-i}g^i - g^k\|^2]\right].
$$

Taking $\rho = \frac{1}{1-\beta}$ and $\rho_1 = \frac{1}{\beta}$ gives

$$
\left[1 + 2\mu\left(\left(\alpha\frac{\beta^2}{1-\beta} + \frac{L\alpha^2}{2}(\frac{\beta}{1-\beta})^2\right)4(1-\beta^{k-1})^2 + \alpha\frac{1}{1-\beta}\right)\right]\mathbb{E}[\|g^k\|^2]
$$

$$
\geq 2\mu\left[\mathbb{E}[f(z^k)] - f^\star\right.
$$

$$
- \left(\alpha\frac{\beta^2}{1-\beta} + \frac{L\alpha^2}{2}(\frac{\beta}{1-\beta})^2\right)2\frac{1-\beta}{1+\beta}\sigma^2 \tag{39}
$$

$$
- \left(\alpha\frac{1}{2(1-\beta)} + \left(\alpha\frac{\beta^2}{1-\beta} + \frac{L\alpha^2}{2}(\frac{\beta}{1-\beta})^2\right)4(1-\beta^k)^2\frac{1}{\beta^2} + \alpha\frac{1}{1-\beta}\right)
$$

$$
\left.\times\mathbb{E}[\|\frac{1}{1-\beta^k}(1-\beta)\sum_{i=1}^{k}\beta^{k-i}g^i - g^k\|^2]\right].
$$

Since

$$
\alpha \leq \frac{1-\beta}{5L}, \tag{40}
$$

(39) gives

$$\left(1 + 8\frac{\mu}{L}\right) \mathbb{E}[\|g^k\|^2]$$

$$\geq 2\mu \Bigg[ \mathbb{E}[f(z^k)] - f^\star$$

$$- \left(\alpha\frac{\beta^2}{1-\beta} + \frac{L\alpha^2}{2}(\frac{\beta}{1-\beta})^2\right) 2\frac{1-\beta}{1+\beta}\sigma^2 \tag{41}$$

$$- \left(\alpha\frac{1}{2(1-\beta)} + \left(\alpha\frac{\beta^2}{1-\beta} + \frac{L\alpha^2}{2}(\frac{\beta}{1-\beta})^2\right) 4(1-\beta^k)^2 \frac{1}{\beta^2} + \alpha\frac{1}{1-\beta}\right)$$

$$\times \mathbb{E}[\|\frac{1}{1-\beta^k}(1-\beta)\sum_{i=1}^{k}\beta^{k-i}g^i - g^k\|^2] \Bigg].$$

Since $\alpha \leq \frac{1-\beta}{5L}$, we have that

$$c_1 = \left(\frac{\sqrt{\beta}}{(1-\sqrt{\beta})^2} + \frac{\sqrt{\beta}}{1-\sqrt{\beta}}\frac{\beta}{1-\beta}\right)\left(\frac{2L^3\alpha^2}{1-\beta} + \frac{18L^2\mu\alpha^2}{(1-\beta)(1+\frac{8\mu}{L})}\right)$$

$$\leq \left(\frac{4\sqrt{\beta}}{(1-\beta)^2} + \frac{2\sqrt{\beta}}{1-\beta}\frac{\beta}{1-\beta}\right)\left(\frac{2L(1-\beta)}{25} + \frac{18\mu(1-\beta)}{25(1+\frac{8\mu}{L})}\right) \tag{42}$$

$$\leq \frac{6\sqrt{\beta}}{25(1-\beta)}\left(2L + \frac{18\mu}{1+\frac{8\mu}{L}}\right)$$

$$\leq \frac{6\sqrt{\beta}}{25(1-\beta)}\left(2L + 18\mu\right)$$

Therefore, by $\alpha \leq \frac{1-\beta}{L\left(3-\beta+2\beta^2+\frac{48\sqrt{\beta}}{25}\frac{2L+18\mu}{L}\right)}$ we have

$$- \alpha + \frac{3-\beta+2\beta^2}{2(1-\beta)}L\alpha^2 + 4c_1\alpha^2$$

$$= -\frac{\alpha}{2} - \frac{\alpha}{2} + \frac{3-\beta+2\beta^2}{2(1-\beta)}L\alpha^2 + \frac{24\sqrt{\beta}}{25(1-\beta)}\left(2L+18\mu\right)\alpha^2 \tag{43}$$

$$\leq -\frac{\alpha}{2}.$$

Combine (43) with (36), we have

$$\mathbb{E}[L^{k+1} - L^k] \leq -\frac{\alpha}{2}\mathbb{E}[\|g^k\|^2] + \left(\frac{\beta^2}{2(1+\beta)}L\alpha^2\sigma^2 + \frac{1}{2}L\alpha^2\sigma^2 + 2c_1\frac{1-\beta}{1+\beta}\alpha^2\sigma^2\right)$$

$$+ \sum_{i=1}^{k-1}(c_{i+1}-c_i)\mathbb{E}[\|x^{k+1-i} - x^{k-i}\|^2] \tag{44}$$

$$+ \left(4c_1\alpha^2 + \frac{L\alpha^2}{(1-\beta)}\right)(1-\beta^k)^2\mathbb{E}[\|\frac{1}{1-\beta^k}(1-\beta)\sum_{i=1}^{k}\beta^{k-i}g^i - g^k\|^2].$$

By combining (44) with (41), we further obtain

$$\mathbb{E}[L^{k+1} - L^k] \leq B_1\mathbb{E}[f(z^k) - f^\star] + B_2$$

$$+ B_3\mathbb{E}[\|\frac{1}{1-\beta^k}(1-\beta)\sum_{i=1}^{k}\beta^{k-i}g^i - g^k\|^2] + \sum_{i=1}^{k-1}(c_{i+1}-c_i)\mathbb{E}[\|x^{k+1-i} - x^{k-i}\|^2],$$

$$\tag{45}$$

where

$$B_1 = -\frac{\alpha}{2}\frac{2\mu}{1+\frac{8\mu}{L}},$$

$$B_2 = \frac{\beta^2}{2(1+\beta)}L\alpha^2\sigma^2 + \frac{1}{2}L\alpha^2\sigma^2 + 2c_1\frac{1-\beta}{1+\beta}\alpha^2\sigma^2$$

$$+ \frac{\alpha}{2}\frac{2\mu\left(\alpha\frac{\beta^2}{1-\beta} + \frac{L\alpha^2}{2}(\frac{\beta}{1-\beta})^2\right)2\frac{1-\beta}{1+\beta}\sigma^2}{1+\frac{8\mu}{L}},$$

$$B_3 = 4c_1\alpha^2 + \frac{L\alpha^2}{(1-\beta)}$$

$$+ \frac{\alpha}{2}\frac{2\mu\left(\alpha\frac{1}{2(1-\beta)} + \left(\alpha\frac{\beta^2}{1-\beta} + \frac{L\alpha^2}{2}(\frac{\beta}{1-\beta})^2\right)4\frac{1}{\beta^2} + \alpha\frac{1}{1-\beta}\right)}{1+\frac{8\mu}{L}}.$$

(46)

From Lemma 2 we know that

$$\mathbb{E}\left[\|\frac{1}{1-\beta^k}(1-\beta)\sum_{i=1}^{k}\beta^{k-i}g^i - g^k\|^2\right] \leq \sum_{i=1}^{k-1}a_{k,i}\,\mathbb{E}[\||x^{i+1}-x^i\||^2],$$

where

$$a_{k,i} = \frac{L^2\beta^{k-i}}{1-\beta^k}\left(k-i+\frac{\beta}{1-\beta}\right).$$

(47)

Putting this into (45) yields

$$\mathbb{E}[L^{k+1}-L^k] \leq B_1\,\mathbb{E}[f(z^k)-f^\star] + B_2$$

$$+ \sum_{i=1}^{k-1}(c_{i+1}-c_i+B_3 a_{k,k-i})\,\mathbb{E}[\||x^{k+1-i}-x^{k-i}\||^2].$$

(48)

In the rest of the proof, we will show that if the constants $c_i$ are chosen such that

$$c_1 = \left(\frac{\sqrt{\beta}}{(1-\sqrt{\beta})^2} + \frac{\sqrt{\beta}}{1-\sqrt{\beta}}\frac{\beta}{1-\beta}\right)\left(\frac{4L^3\alpha^2}{1-\beta} + \frac{30L^2\mu\alpha^2}{(1-\beta)(1+\frac{8\mu}{L})}\right),$$

(49)

and

$$c_{i+1}-c_i+B_3 2L^2\beta^{k-i}\left(k-i+\frac{\beta}{1-\beta}\right) = B_1 c_i, \qquad \forall i \geq 1.$$

(50)

Then, we have $c_i > 0$ for all $i \geq 1$ and

$$c_{i+1}-c_i+B_3 a_{k,k-i} \leq B_1 c_i, \qquad \forall i \geq 1.$$

(51)

And therefore, we will have the desired result:

$$\mathbb{E}[L^{k+1}-L^k] \leq B_1\,\mathbb{E}[f(z^k)-f^\star] + B_2 + B_1\sum_{i=1}^{k-1}c_i\,\mathbb{E}[\||x^{k+1-i}-x^{k-i}\||^2]$$

$$= -\frac{\alpha\mu}{1+\frac{8\mu}{L}}\,\mathbb{E}[L^k] + \frac{\beta^2}{2(1+\beta)}L\alpha^2\sigma^2 + \frac{1}{2}L\alpha^2\sigma^2 + 2c_1\frac{1-\beta}{1+\beta}\alpha^2\sigma^2$$

$$+ \frac{\beta^2 + \frac{L\alpha}{2}\frac{\beta^2}{1-\beta}}{1+\frac{8\mu}{L}}\frac{2}{1+\beta}\mu\alpha^2\sigma^2.$$

First of all. by $k \geq \frac{\log 0.5}{\log \beta}$, we know that $\beta^k \leq \frac{1}{2}$, and (47) gives

$$a_{k,k-i} \leq 2L^2\beta^i\left(i+\frac{\beta}{1-\beta}\right).$$

Therefore, in order for (51) to hold, it suffices to set

$$c_{i+1} - c_i + B_3 2L^2 \beta^{k-i} \left( k - i + \frac{\beta}{1-\beta} \right) = B_1 c_i \qquad \forall i \geq 1.$$

This is exactly (50).

On the other hand, (50) is also equivalent to

$$\frac{c_{i+1}}{(1+B_1)^{i+1}} - \frac{c_i}{(1+B_1)^i} = -\frac{2L^2 B_3}{(1+B_1)^{i+1}} \beta^i \left( i + \frac{\beta}{1-\beta} \right), \qquad \forall i \geq 1.$$

Therefore, in order to have $c_i > 0$ for all $i \geq 1$, we can set

$$c_1 \geq 2L^2 B_3 \sum_{i=1}^{\infty} \left( \frac{\beta}{1+B_1} \right)^i \left( i + \frac{\beta}{1-\beta} \right). \tag{52}$$

Since $\beta \leq \sqrt{\beta} \leq 1 + B_1 = 1 - \alpha\mu \frac{1}{1+\frac{8\mu}{L}}$ and

$$\sum_{i=1}^{j} iq^i = \frac{1}{1-q} \left( \frac{q(1-q^j)}{1-q} - jq^{j+1} \right),$$

for any $q \in (0,1)$, (52) is equivalent to

$$c_1 \geq 2L^2 B_3 \left( \frac{\frac{\beta}{1+B_1}}{(1-\frac{\beta}{1+B_1})^2} + \frac{\frac{\beta}{1+B_1}}{1-\frac{\beta}{1+B_1}} \frac{\beta}{1-\beta} \right). \tag{53}$$

Recall from (46) that

$$B_3 = 4c_1\alpha^2 + \frac{L\alpha^2}{(1-\beta)}$$

$$+ \frac{\alpha}{2} \frac{2\mu \left( \alpha \frac{1}{2(1-\beta)} + \left( \alpha \frac{\beta^2}{1-\beta} + \frac{L\alpha^2}{2}(\frac{\beta}{1-\beta})^2 \right) 4\frac{1}{\beta^2} + \alpha \frac{1}{1-\beta} \right)}{1+\frac{8\mu}{L}}$$

$$= \left( 4c_1\alpha^2 + \frac{L\alpha^2}{(1-\beta)} \right) + \frac{\alpha}{2} \frac{2\mu \left( \alpha \frac{11}{2(1-\beta)} + 2L\alpha^2(\frac{1}{1-\beta})^2 \right)}{1+\frac{8\mu}{L}}.$$

Since $\alpha \leq \frac{1-\beta}{L}$, we further have

$$B_3 \leq \left( 4c_1\alpha^2 + \frac{L\alpha^2}{(1-\beta)} \right) + \frac{\alpha}{2} \frac{2\mu \left( \alpha \frac{15}{2(1-\beta)} \right)}{1+\frac{8\mu}{L}}.$$

Since $B_1 = -\frac{\alpha\mu}{1+\frac{8\mu}{L}}$ and $\alpha \leq \frac{1-\beta}{5L}$, it can be verified that $\frac{\beta}{1+B_1} \leq \sqrt{\beta}$ for all $\beta \in [0,1)$ and $\mu \leq L$.
Therefore,

$$\frac{\frac{\beta}{1+B_1}}{(1-\frac{\beta}{1+B_1})^2} + \frac{\frac{\beta}{1+B_1}}{1-\frac{\beta}{1+B_1}} \frac{\beta}{1-\beta} \leq \frac{\sqrt{\beta}}{(1-\sqrt{\beta})^2} + \frac{\sqrt{\beta}}{1-\sqrt{\beta}} \frac{\beta}{1-\beta}.$$

As a result, in order to have (53), it suffices to set

$$c_1 \geq 2L^2 \left( \frac{\sqrt{\beta}}{(1-\sqrt{\beta})^2} + \frac{\sqrt{\beta}}{1-\sqrt{\beta}} \frac{\beta}{1-\beta} \right) \left( 4c_1\alpha^2 + \frac{L\alpha^2}{(1-\beta)} + \frac{\alpha}{2} \frac{2\mu \left( \alpha \frac{15}{2(1-\beta)} \right)}{1+\frac{8\mu}{L}} \right), \tag{54}$$

Since $\alpha \leq \frac{1-\beta}{5L}$, we have

$$1 - 8\alpha^2 L^2 \left( \frac{\sqrt{\beta}}{(1-\sqrt{\beta})^2} + \frac{\sqrt{\beta}}{1-\sqrt{\beta}} \frac{\beta}{1-\beta} \right) \geq \frac{1}{2},$$

(54) in turn just requires

$$c_1 = \left( \frac{\sqrt{\beta}}{(1-\sqrt{\beta})^2} + \frac{\sqrt{\beta}}{1-\sqrt{\beta}} \frac{\beta}{1-\beta} \right) \left( \frac{4L^3\alpha^2}{1-\beta} + \frac{30L^2\mu\alpha^2}{(1-\beta)(1+\frac{8\mu}{L})} \right),$$

which is exactly our choice of $c_1$ as in (49).

## B.5 Proof of Theorem 2

From Proposition 2 we know that for all $k \geq k_0 = \lfloor \frac{\log 0.5}{\log \beta} \rfloor$,

$$\mathbb{E}[L^{k+1} - L^k] \leq -\frac{\alpha\mu}{1 + \frac{8\mu}{L}} \mathbb{E}[L^k] + \frac{1 + \beta + \beta^2}{2(1 + \beta)} L\alpha^2\sigma^2 + \frac{1 - \beta}{1 + \beta} 2c_1\alpha^2\sigma^2$$

$$+ \frac{\beta^2 + \frac{L\alpha}{2}\frac{\beta^2}{1-\beta}}{(1 + \frac{8\mu}{L})(1 + \beta)} 2\mu\alpha^2\sigma^2.$$

Rearranging gives

$$\mathbb{E}[L^{k+1}] \leq \left(1 - \frac{\alpha\mu}{1 + \frac{8\mu}{L}}\right) \mathbb{E}[L^k] + \frac{1 + \beta + \beta^2}{2(1 + \beta)} L\alpha^2\sigma^2 + \frac{1 - \beta}{1 + \beta} 2c_1\alpha^2\sigma^2$$

$$+ \frac{\beta^2 + \frac{L\alpha}{2}\frac{\beta^2}{1-\beta}}{(1 + \frac{8\mu}{L})(1 + \beta)} 2\mu\alpha^2\sigma^2$$

$$\leq \left(1 - \frac{\alpha\mu}{1 + \frac{8\mu}{L}}\right) \mathbb{E}[L^k] + \frac{1 + \beta + \beta^2}{2(1 + \beta)} L\alpha^2\sigma^2 + \frac{1 - \beta}{1 + \beta} 2c_1\alpha^2\sigma^2$$

$$+ \frac{\beta^2 + \frac{L\alpha}{10}\beta^2}{1 + \frac{8\mu}{L}} \frac{2}{1 + \beta} \mu\alpha^2\sigma^2,$$

where we have applied $\alpha \leq \frac{1-\beta}{5L}$ in the last step. Therefore,

$$\mathbb{E}[L^{k+1}] - \frac{1}{\frac{\alpha\mu}{1+\frac{8\mu}{L}}} \left( \frac{1 + \beta + \beta^2}{2(1 + \beta)} L\alpha^2\sigma^2 + \frac{1 - \beta}{1 + \beta} 2c_1\alpha^2\sigma^2 + \frac{\beta^2 + \frac{L\alpha}{10}\beta^2}{1 + \frac{8\mu}{L}} \frac{2}{1 + \beta} \mu\alpha^2\sigma^2 \right)$$

$$\leq \left(1 - \frac{\alpha\mu}{1 + \frac{8\mu}{L}}\right)$$

$$\times \left( \mathbb{E}[L^k] - \frac{1}{\frac{\alpha\mu}{1+\frac{8\mu}{L}}} \left( \frac{1 + \beta + \beta^2}{2(1 + \beta)} L\alpha^2\sigma^2 + \frac{1 - \beta}{1 + \beta} 2c_1\alpha^2\sigma^2 + \frac{\beta^2 + \frac{L\alpha}{10}\beta^2}{1 + \frac{8\mu}{L}} \frac{2}{1 + \beta} \mu\alpha^2\sigma^2 \right) \right).$$

This immediately yields

$$\mathbb{E}[L^k]$$

$$\leq \left(1 - \frac{\alpha\mu}{1 + \frac{8\mu}{L}}\right)^{k-k_0}$$

$$\times \left( \mathbb{E}[L^{k_0}] - \frac{1}{\frac{\alpha\mu}{1+\frac{8\mu}{L}}} \left( \frac{1 + \beta + \beta^2}{2(1 + \beta)} L\alpha^2\sigma^2 + \frac{1 - \beta}{1 + \beta} 2c_1\alpha^2\sigma^2 + \frac{\beta^2 + \frac{L\alpha}{10}\beta^2}{1 + \frac{8\mu}{L}} \frac{2}{1 + \beta} \mu\alpha^2\sigma^2 \right) \right)$$

$$+ \frac{1}{\frac{\alpha\mu}{1+\frac{8\mu}{L}}} \left( \frac{1 + \beta + \beta^2}{2(1 + \beta)} L\alpha^2\sigma^2 + \frac{1 - \beta}{1 + \beta} 2c_1\alpha^2\sigma^2 + \frac{\beta^2 + \frac{L\alpha}{10}\beta^2}{1 + \frac{8\mu}{L}} \frac{2}{1 + \beta} \mu\alpha^2\sigma^2 \right)$$

$$\leq \left(1 - \frac{\alpha\mu}{1 + \frac{8\mu}{L}}\right)^{k-k_0} \mathbb{E}[L^{k_0}]$$

$$+ \left(1 + \frac{8\mu}{L}\right) \left( \frac{1 + \beta + \beta^2}{4(1 + \beta)} \frac{L}{\mu}\alpha\sigma^2 + \frac{1 - \beta}{1 + \beta} \frac{2c_1}{\mu}\alpha\sigma^2 + \frac{\beta^2 + \frac{L\alpha}{10}\beta^2}{1 + \frac{8\mu}{L}} \frac{2}{1 + \beta}\alpha\sigma^2 \right).$$

By $c_i \geq 0$ for all $i \geq 1$ and (42), we conclude that

$$\mathbb{E}[f(z^k) - f^*]$$

$$\leq \left(1 - \frac{\alpha\mu}{1 + \frac{8\mu}{L}}\right)^{k-k_0} \mathbb{E}[L^{k_0}]$$

$$+ \left(1 + \frac{8\mu}{L}\right)\left(\frac{1 + \beta + \beta^2}{2(1+\beta)}\frac{L}{\mu}\alpha\sigma^2 + \frac{1}{1+\beta}\frac{12\sqrt{\beta}}{25}\frac{2L + 18\mu}{\mu}\alpha\sigma^2 + \frac{\beta^2 + \frac{L\alpha}{10}\beta^2}{1 + \frac{8\mu}{L}}\frac{2}{1+\beta}\alpha\sigma^2\right)$$

$$= \mathcal{O}\left((1 - \alpha\mu)^{k-k_0} + \frac{L}{\mu}\alpha\sigma^2\right).$$

### B.6  Proof of Corollary 1

In fact, by (6) we can express $x^k$ as a convex combination of $\{z^i\}_{i=1}^k$:

$$x^k = (1 - \beta)\sum_{i=2}^{k}\beta^{k-i}z^i + \beta^{k-1}z^1.$$

The desired result follows directly from the convexity of $f$ and Theorem 2.

## C  Generalizations of Lemmas 1, 2, and 3 for Multistage SGDM

In order to establish the convergence of Multistage SGDM(Algorithm 1), we need to generalize the Lemmas 1 and 2 , which play a key role in the convergence of SGDM in (2).

### C.1  Generalization of Lemma 1 for Multistage SGDM

**Lemma 4.** *Under the assumptions of Theorem 3, the variance of update vector $m^k$ in Algorithm 1 satisfies*

$$\frac{1}{1 - \beta_1}\mathbb{E}[\|m^k - \sum_{i=1}^{k}b_{k,i}g^i\|^2] \leq 2\sigma^2,$$

*where* $b_{k,i} = \left(1 - \beta(i)\right)\prod_{j=i+1}^{k}\beta(j)$.

*Proof.* To begin with, let us express $m^k$ by the past stochastic gradients:

$$m^k = \beta(k)m^{k-1} + \left(1 - \beta(k)\right)\tilde{g}^k$$

$$= \beta(k)\beta(k-1)m^{k-2} + \beta(k)\left(1 - \beta(k-1)\right)\tilde{g}^{k-1}$$

$$+ \cdots + \left(1 - \beta(k)\right)\tilde{g}^k$$

$$= ...$$

$$= \prod_{i=1}^{k}\beta(i)m^0 + \prod_{i=2}^{k}\beta(i)\left(1 - \beta(1)\right)\tilde{g}^1 \tag{55}$$

$$+ \cdots + \left(1 - \beta(k)\right)\tilde{g}^k$$

$$= \sum_{i=1}^{k}b_{k,i}\tilde{g}^i,$$

where we have applied $m^0 = 0$ and defined

$$b_{k,i} = \left(1 - \beta(i)\right)\prod_{j=i+1}^{k}\beta(j) \tag{56}$$

in the last step.

It can be verified that the sum of weights is

$$\sum_{i=1}^{k} b_{k,i} = 1 - \prod_{i=1}^{k} \beta(i).$$ (57)

As a result, by applying Assumption 1 we have

$$\mathbb{E}[\|m^k - \sum_{i=1}^{k} b_{k,i} g^i\|^2] = \mathbb{E}[\|\sum_{i=1}^{k} b_{k,i}(\tilde{g}^i - g^i)\|^2] \leq \sum_{i=1}^{k} b_{k,i}^2 \sigma^2.$$

Note that by setting $k = T_1 + \cdots + T_{n_k} + r_k$, we have

$$b_{k,i} = \begin{cases} \beta_{n_k+1}^{r_k} \beta_{n_k}^{T_{n_k}} \ldots \beta_2^{T_2} (1 - \beta_1) \beta_1^{T_1 - i}, 1 \leq i \leq T_1, \\ \beta_{n_k+1}^{r_k} \beta_{n_k}^{T_{n_k}} \ldots \beta_3^{T_3} (1 - \beta_2) \beta_1^{T_1 + T_2 - i}, T_1 + 1 \leq i \leq T_1 + T_2, \\ \ldots\ldots \\ (1 - \beta_{n_k+1}) \beta_1^{T_1 + \cdots + T_{n_k} + r_k - i}, \sum_{l=1}^{n_k} T_l + 1 \leq i \leq \sum_{l=1}^{n_k} T_l + r_k. \end{cases}$$

Therefore,

$$\mathbb{E}[\|m^k - \sum_{i=1}^{k} b_{k,i} g^i\|^2] \leq (\beta_{n_k+1}^{r_k} \beta_{n_k}^{T_{n_k}} \ldots \beta_2^{T_2})^2 \frac{1 - \beta_1}{1 + \beta_1} (1 - \beta_1^{2T_1}) \sigma^2$$

$$+ (\beta_{n_k+1}^{r_k} \beta_{n_k}^{T_{n_k}} \ldots \beta_3^{T_3})^2 \frac{1 - \beta_2}{1 + \beta_2} (1 - \beta_2^{2T_2}) \sigma^2$$

$$+ \ldots$$

$$+ (\beta_{n_k+1}^{r_k})^2 \frac{1 - \beta_{n_k}}{1 + \beta_{n_k}} (1 - \beta_{n_k}^{2T_{n_k}}) \sigma^2$$

$$+ \frac{1 - \beta_{n_k+1}}{1 + \beta_{n_k+1}} (1 - \beta_{n_k+1}^{2r_k}) \sigma^2.$$

Since for any $l \in [1, n]$, we have

$$(\beta_l^{T_l})^2 \leq \frac{1}{2},$$
$$1 - \beta_l \leq 1 - \beta_1,$$
$$1 + \beta_l \geq \frac{3}{2},$$
$$1 - \beta_l^{2T_l} < 1.$$

Therefore,

$$\frac{1}{1 - \beta_1} \mathbb{E}[\|m^k - \sum_{i=1}^{k} b_{k,i} g^i\|^2] \leq (\beta_{n_k+1}^{r_k})^2 (\frac{1}{2})^{n_k - 1} \frac{2}{3} \cdot \frac{1 - \beta_1}{1 - \beta_1} \sigma^2$$

$$+ (\beta_{n_k+1}^{r_k})^2 (\frac{1}{2})^{n_k - 2} \frac{2}{3} \cdot \frac{1 - \beta_2}{1 - \beta_1} \sigma^2$$

$$+ \ldots$$

$$+ (\beta_{n_k+1}^{r_k})^2 (\frac{1}{2})^0 \frac{2}{3} \cdot \frac{1 - \beta_{n_k}}{1 - \beta_1} \sigma^2$$

$$+ \frac{2}{3} \cdot \frac{1 - \beta_{n_k+1}}{1 - \beta_1} \sigma^2$$

$$\leq 2\sigma^2.$$

$\square$

**Lemma 5.** *Under the assumptions of Theorem 3, the update vector $m^{k-1}$ in Algorithm 1 satisfies*

$$\frac{1}{1-\beta(k)}\mathbb{E}[\|m^{k-1} - \sum_{i=1}^{k-1} b_{k-1,i}g^i\|^2] \le 24\frac{\beta_1}{\sqrt{\beta_n + \beta_n^2}}\sigma^2.$$

*Proof.* By setting $k-1 = T_1 + \cdots + T_{n_{k-1}} + r_{k-1}$, we have

$$
\begin{aligned}
\mathbb{E}[\|m^{k-1} - \sum_{i=1}^{k-1} b_{k-1,i}g^i\|^2] &\le (\beta_{n_{k-1}+1}^{r_{k-1}}\beta_{n_{k-1}}^{T_{n_{k-1}}}...\beta_2^{T_2})^2\frac{1-\beta_1}{1+\beta_1}(1-\beta_1^{2T_1})\sigma^2 \\
&+ (\beta_{n_{k-1}+1}^{r_{k-1}}\beta_{n_{k-1}}^{T_{n_{k-1}}}...\beta_3^{T_3})^2\frac{1-\beta_2}{1+\beta_2}(1-\beta_2^{2T_2})\sigma^2 \\
&+ \ldots \\
&+ (\beta_{n_{k-1}+1}^{r_{k-1}})^2\frac{1-\beta_{n_{k-1}}}{1+\beta_{n_{k-1}}}(1-\beta_{n_{k-1}}^{2T_{n_{k-1}}})\sigma^2 \\
&+ \frac{1-\beta_{n_{k-1}+1}}{1+\beta_{n_{k-1}+1}}(1-\beta_{n_{k-1}+1}^{2r_{k-1}})\sigma^2.
\end{aligned}
$$

Similar as before, we have

$$
\begin{aligned}
\frac{1}{1-\beta(k)}\mathbb{E}[\|m^{k-1} - \sum_{i=1}^{k} b_{k-1,i}g^i\|^2] &\le (\beta_{n_{k-1}+1}^{r_{k-1}})^2(\frac{1}{2})^{n_{k-1}-1}\frac{2}{3}\cdot\frac{1-\beta_1}{1-\beta(k)}\sigma^2 \\
&+ (\beta_{n_{k-1}+1}^{r_{k-1}})^2(\frac{1}{2})^{n_{k-1}-2}\frac{2}{3}\cdot\frac{1-\beta_1}{1-\beta(k)}\sigma^2 \\
&+ \ldots \\
&+ (\beta_{n_{k-1}+1}^{r_{k-1}})^2(\frac{1}{2})^0\frac{2}{3}\cdot\frac{1-\beta_1}{1-\beta(k)}\sigma^2 \\
&+ \frac{2}{3}\cdot\frac{1-\beta_1}{1-\beta(k)}\sigma^2 \\
&\le 2\frac{1-\beta_1}{1-\beta(k)}\sigma^2.
\end{aligned}
$$

Finally, by applying

$$\frac{1-\beta_1}{1-\beta_n} \le 12\frac{\beta_1}{\sqrt{\beta_n + \beta_n^2}},$$

we arrive at

$$\frac{1}{1-\beta(k)}\mathbb{E}[\|m^{k-1} - \sum_{i=1}^{k} b_{k-1,i}g^i\|^2] \le 24\frac{\beta_1}{\sqrt{\beta_n + \beta_n^2}}\sigma^2.$$

$\square$

## C.2 Generalization of Lemma 3 for Multistage SGDM

**Lemma 6.** *$z^k$ defined in (8) satisfies*

$$z^{k+1} - z^k = -\alpha(k)\tilde{g}^k,$$

*where $\alpha(k)$ is the stepsize applied at the kth step.*

*Proof.* Recall that the auxiliary sequence $z^k$ is defined by

$$z^k = x^k - A_1 m^{k-1},$$

where $A_1 \equiv \frac{\alpha_i \beta_i}{1 - \beta_i}$ and $\alpha_i, \beta_i$ are the stepsize and momentum weight at the $i$th stage, respectively. Therefore, we also have

$$A_1 \equiv \frac{\alpha(k)\beta(k)}{1 - \beta(k)},$$

where $\alpha(k), \beta(k)$ are the stepsize and momentum weight applied at the $k$th step. Using this, we obtain

$$
\begin{aligned}
z^{k+1} - z^k &= x^{k+1} - x^k - A_1(m^k - m^{k-1}) \\
&= -\alpha(k)m^k - A_1(1 - \beta(k))(\tilde{g}^k - m^{k-1}) \\
&= -\alpha(k)m^k - \alpha(k)\beta(k)(\tilde{g}^k - m^{k-1}) \\
&= \alpha(k)(\beta(k)m^{k-1} - m^k) - \alpha(k)\beta(k)\tilde{g}^k \\
&= -\alpha(k)\tilde{g}^k.
\end{aligned}
$$

$\square$

### C.3 Generalization of Lemma 2 for Multistage SGDM

**Lemma 7.** *In Multistage SGDM(Algorithm 1), assume that the momentum weights at $n$ stages satisfy $\beta_1 \leq \beta_2 \leq ... \leq \beta_n$. Then, we have*

$$
\mathbb{E}\left[\left\|\frac{1}{1 - \prod_{i=1}^k \beta(i)} \sum_{i=1}^k b_{k,i}g^i - g^k\right\|^2\right] \leq \sum_{i=1}^{k-1} a_{k,i} \mathbb{E}[\|x^{j+1} - x^j\|^2],
$$

*where $b_{k,i} = \left(1 - \beta(i)\right)\prod_{j=i+1}^k \beta(j)$ and $\beta(i)$ is the momentum weight applied at the $i$th iteration, and*

$$
a_{k,i} = \frac{L^2 \beta^{k-i}(k)}{1 - \prod_{i=1}^k \beta(i)}\left(k - i + \frac{\beta(k)}{1 - \beta(k)}\right). \tag{58}
$$

*Proof.* By By (55), (56) and (57), we can compute that

$$
\mathbb{E}\left[\left\|\frac{1}{1 - \prod_{i=1}^k \beta(i)} \sum_{i=1}^k b_{k,i}g^i - g^k\right\|^2\right]
$$

$$
= \mathbb{E}\left\|\frac{1}{1 - \prod_{j=1}^k \beta(j)} \sum_{i=1}^k b_{k,i}(g^i - g^k)\right\|^2
$$

$$
= \left(\frac{1}{1 - \prod_{j=1}^k \beta(j)}\right)^2 \sum_{i,j=1}^k b_{k,i}b_{k,j} \mathbb{E}\langle(g^k - g^i), (g^k - g^j)\rangle
$$

$$
\leq \left(\frac{1}{1 - \prod_{j=1}^k \beta(j)}\right)^2 \sum_{i,j=1}^k b_{k,i}b_{k,j}(\frac{1}{2}\mathbb{E}\|g^k - g^i\|^2
$$

$$
+ \frac{1}{2}\mathbb{E}\|g^k - g^j\|^2)
$$

$$
= \left(\frac{1}{1 - \prod_{j=1}^k \beta(j)}\right) \sum_{j=1}^k b_{k,j} \mathbb{E}\|g^k - g^j\|^2
$$

$$
\leq \left(\frac{1}{1 - \prod_{j=1}^k \beta(j)}\right) \sum_{j=1}^k b_{k,j}(k - j)\sum_{i=j}^{k-1} L^2 \mathbb{E}\|x^{i+1} - x^i\|^2,
$$

where we have used (57) in the first and third equality, and Cauchy-Schwarz in the first inequality. In the last inequality, we have applied the triangle inequality and the $L-$smoothness of $f$.

Consequently, we have

$$
\mathbb{E}\left\|\frac{1}{1 - \prod_{j=1}^{k} \beta(j)} \sum_{i=1}^{k} b_{k,i} g^i - g^k\right\|^2
$$

$$
\leq \left(\frac{1}{1 - \prod_{j=1}^{k} \beta(j)}\right) \sum_{j=1}^{k} b_{k,j}(k-j) \sum_{i=j}^{k-1} L^2 \, \mathbb{E}\, \|x^{i+1} - x^i\|^2
$$

$$
= \left(\frac{1}{1 - \prod_{j=1}^{k} \beta(j)}\right) \sum_{i=1}^{k-1} \sum_{j=1}^{i} b_{k,j}(k-j)L^2 \, \mathbb{E}\, \|x^{i+1} - x^i\|^2
$$

$$
= \sum_{i=1}^{k-1} d_{k,i} \, \mathbb{E}[\|x^{i+1} - x^i\|^2],
$$

where in the last step we have defined

$$
d_{k,i} = \left(\frac{L^2}{1 - \prod_{j=1}^{k} \beta(j)}\right) \sum_{j=1}^{i} (k-j)b_{k,j}. \tag{59}
$$

In the Proposition 5 below, we shall see that $d_{k,i} \leq a_{k,i}$ for all $i \leq k-1$, where $a_{k,i}$ is defined in (58). Therefore,

$$
\mathbb{E}[\|\frac{1}{1 - \prod_{i=1}^{k} \beta(i)} \sum_{i=1}^{k} b_{k,i} g^i - g^k\|^2] \leq \sum_{i=1}^{k-1} a_{k,i} \, \mathbb{E}[\|x^{j+1} - x^j\|^2],
$$

and the proof will be complete.

$\square$

**Proposition 5.** $d_{k,i}$ *defined in* (59) *and* $a_{k,i}$ *defined in* (58) *satisfy*

$$
d_{k,i} \leq a_{k,i} \quad \textit{for all} \quad i \leq k-1.
$$

*Proof.* We aim to show that $d_{k,i} \leq a_{k,i}$ for all $i \leq k-1$. Or equivalently, $d_{k,j} \leq a_{k,j}$ for all $j \leq k-1$.

In order to show $d_{k,j} \leq a_{k,j}$, we just need to show that

$$
\sum_{i=1}^{j} (k-i)b_{k,i} \leq \beta^{k-j}(k) \left(k - j + \frac{\beta(k)}{1 - \beta(k)}\right), \tag{60}
$$

where

$$
b_{k,i} = \left(1 - \beta(i)\right) \prod_{j=i+1}^{k} \beta(j).
$$

Let $k = T_1 + T_2 + \cdots + T_{n_k} + r_k$, where $0 \leq n_k \leq n-1$. If $n_k < n-1$, then $0 \leq r_k \leq T_{n_k+1} - 1$. If $n_k = n-1$, then $0 \leq r_k \leq T_{n_k+1} = T_n$.

Since $j \leq k-1$, we have $j = T_1 + \cdots + T_{n_j} + r_j$, where $0 \leq n_j \leq n_k$.

Now, let us compute the left hand side of (60) explicitly.

$$
\sum_{i=1}^{j} (k-i)b_{k,i}
$$

$$
= \left(\sum_{i=1}^{T_1} + \sum_{i=T_1+1}^{T_1+T_2} + \cdots + \sum_{i=T_1+\cdots+T_{n_j}+1}^{T_1+\cdots+T_{n_j}+r_j}\right)(k-i)b_{k,i}.
$$

Notice that

$$
b_{k,i} = \begin{cases}
\beta_{n_k+1}^{r_k}\beta_{n_k}^{T_{n_k}}\cdots\beta_2^{T_2}(1-\beta_1)\beta_1^{T_1-i}, & 1 \le i \le T_1, \\
\beta_{n_k+1}^{r_k}\beta_{n_k}^{T_{n_k}}\cdots\beta_3^{T_3}(1-\beta_2)\beta_1^{T_1+T_2-i}, & \\
& T_1+1 \le i \le T_1+T_2, \\
\cdots\cdots & \\
(1-\beta_{n_k+1})\beta_1^{T_1+\cdots+T_{n_k}+r_k-i}, & \\
& \sum_{l=1}^{n_k} T_l + 1 \le i \le \sum_{l=1}^{n_k} T_l + r_k.
\end{cases}
$$

As a result, we have

$$
\begin{aligned}
& \sum_{i=1}^{j}(k-i)b_{k,i} \\
& = \left( \sum_{i=1}^{T_1} + \sum_{i=T_1+1}^{T_1+T_2} + \cdots + \sum_{i=T_1+\cdots+T_{n_j}+1}^{T_1+\cdots+T_{n_j}+r_j} \right)(k-i)b_{k,i} \\
& \le \beta_{n_k+1}^{r_k}\beta_{n_k}^{T_{n_k}}\cdots\beta_2^{T_2}(1-\beta_1)\sum_{i=1}^{T_1}\beta_1^{T_1-i}(k-i) \\
& \quad + \beta_{n_k+1}^{r_k}\beta_{n_k}^{T_{n_k}}\cdots\beta_3^{T_3}(1-\beta_2)\sum_{i=T_1+1}^{T_1+T_2}\beta_2^{T_1+T_2-i}(k-i) \\
& \quad + \ldots \\
& \quad + \beta_{n_k+1}^{r_k}\beta_{n_k}^{T_{n_k}}...\beta_{n_j+1}^{T_{n_j+1}}(1-\beta_{n_j}) \\
& \qquad \sum_{i=T_1+\cdots+T_{n_j-1}+1}^{T_1+\cdots+T_{n_j}}\beta_{n_l}^{T_1+\cdots+T_{n_j}-i}(k-i) \\
& \quad + \beta_{n_k+1}^{r_k}\beta_{n_k}^{T_{n_k}}...\beta_{n_j+2}^{T_{n_j}+2}(1-\beta_{n_j+1}) \\
& \qquad \sum_{i=T_1+\cdots+T_{n_j}+1}^{T_1+\cdots+T_{n_j}+r_j}\beta_{n_j+1}^{T_1+\cdots+T_{n_j}+r_j-i}(k-i),
\end{aligned} \tag{61}
$$

where we have applied $r_j \le T_{n_j+1}$ if $n_j < n_k$ and $r_j \le r_k$ if $n_j = n_k$ in the last term. Since

$$
\begin{aligned}
\sum_{i=1}^{l}\beta^{k-i}(k-i) = {} & \beta^k\left(-\frac{k-1}{1-\beta}-\frac{1}{(1-\beta)^2}\right) \\
& + \beta^{k-l}\left(\frac{k-l}{1-\beta}+\frac{\beta}{(1-\beta)^2}\right).
\end{aligned}
$$

we have

$$\sum_{i=1}^{T_1} \beta_1^{T_1-i}(k-i) = \beta_1^{T_1-k} \sum_{i=1}^{T_1} \beta_1^{k-i}(k-i)$$

$$= \beta_1^{T_1} \left( -\frac{k-1}{1-\beta_1} - \frac{1}{(1-\beta_1)^2} \right)$$

$$+ \left( \frac{k-T_1}{1-\beta_1} + \frac{\beta_1}{(1-\beta_1)^2} \right),$$

$$\sum_{i=T_1+1}^{T_1+T_2} \beta_2^{T_1+T_2-i}(k-i) = \sum_{i=1}^{T_2} \beta_2^{T_2-i}(k-T_1-i)$$

$$= \beta_2^{T_1+T_2-k} \sum_{i=1}^{T_2} \beta_2^{k-T_1-i}(k-T_1-i)$$

$$= \beta_2^{T_2} \left( -\frac{k-T_1-1}{1-\beta_2} - \frac{1}{(1-\beta_2)^2} \right)$$

$$+ \left( \frac{k-T_1-T_2}{1-\beta_2} + \frac{\beta_2}{(1-\beta_2)^2} \right).$$

And that in general

$$\sum_{i=T_1+\cdots+T_{n_j}+1}^{T_1+\cdots+T_{n_j}+r_j} \beta_{n_j+1}^{T_1+\cdots+T_{n_j}+r_j-i}(k-i)$$

$$= \sum_{i=1}^{r_j} \beta_{n_j+1}^{r_j-i}(k-T_1-\cdots-T_{n_j}-i)$$

$$= \beta_{n_j+1}^{T_1+\cdots+T_{n_j}+r_j-k} \sum_{i=1}^{r_j} \beta_{n_j+1}^{k-T_1-\cdots-T_{n_j}-i}(k-T_1-\cdots-T_{n_j}-i)$$

$$= \beta_{n_j+1}^{r_j} \left( -\frac{k-T_1-\cdots-T_{n_j}-1}{1-\beta_{n_j+1}} - \frac{1}{(1-\beta_{n_j+1})^2} \right)$$

$$+ \left( \frac{k-T_1-\cdots-T_{n_j}-r_j}{1-\beta_{n_j+1}} + \frac{\beta_{n_j+1}}{(1-\beta_{n_j+1})^2} \right).$$

By applying these equalities on (61), we have

$$
\sum_{i=1}^{j}(k-i)b_{k,i}
$$

$$
= \beta_{n_k+1}^{r_k}\beta_{n_k}^{T_{n_k}}...\beta_2^{T_2}\left(\beta_1^{T_1}\left(-(k-1)-\frac{1}{1-\beta_1}\right)+\left((k-T_1)+\frac{\beta_1}{1-\beta_1}\right)\right)
$$

$$
+ \beta_{n_k+1}^{r_k}\beta_{n_k}^{T_{n_k}}...\beta_3^{T_3}\left(\beta_2^{T_2}\left(-(k-T_1-1)-\frac{1}{1-\beta_2}\right)\right.
$$

$$
\left.+\left((k-T_1-T_2)+\frac{\beta_2}{1-\beta_2}\right)\right)
$$

$$
+ \ldots
$$

$$
+ \beta_{n_k+1}^{r_k}\beta_{n_k}^{T_{n_k}}...\beta_{n_j+1}^{T_{n_j+1}}
$$

$$
\left(\beta_{n_j}^{T_{n_j}}\left(-(k-T_1-\cdots-T_{n_j-1}-1)-\frac{1}{1-\beta_{n_j}}\right)\right.
$$

$$
\left.+\left((k-T_1-\cdots-T_{n_j})+\frac{\beta_{n_j}}{1-\beta_{n_j}}\right)\right)
$$

$$
+ \beta_{n_k+1}^{r_k}\beta_{n_k}^{T_{n_k}}...\beta_{n_j+2}^{T_{n_j+2}}
$$

$$
\left(\beta_{n_j+1}^{r_j}\left(-(k-T_1-\cdots-T_{n_j}-1)-\frac{1}{1-\beta_{n_j+1}}\right)\right.
$$

$$
\left.+\left((k-T_1-T_{n_j}-r_j)+\frac{\beta_{n_j+1}}{1-\beta_{n_j+1}}\right)\right).
$$

This yields

$$
\sum_{i=1}^{j}(k-i)b_{k,i} = \beta_{n_k+1}^{r_k}\beta_{n_k}^{T_{n_k}}...\beta_2^{T_2}\beta_1^{T_1}\left(-(k-1)-\frac{1}{1-\beta_1}\right)
$$

$$
+ \beta_{n_k+1}^{r_k}\beta_{n_k}^{T_{n_k}}...\beta_2^{T_2}\left(\frac{\beta_1}{1-\beta_1}+1-\frac{1}{1-\beta_2}\right)
$$

$$
+ \beta_{n_k+1}^{r_k}\beta_{n_k}^{T_{n_k}}...\beta_3^{T_3}\left(\frac{\beta_2}{1-\beta_2}+1-\frac{1}{1-\beta_3}\right)
$$

$$
+ \ldots
$$

$$
+ \beta_{n_k+1}^{r_k}\beta_{n_k}^{T_{n_k}}...\beta_{n_j}^{T_{n_j}}\left(\frac{\beta_{n_j-1}}{1-\beta_{n_j-1}}+1-\frac{1}{1-\beta_{n_j}}\right)
$$

$$
+ \beta_{n_k+1}^{r_k}\beta_{n_k}^{T_{n_k}}...\beta_{n_j+1}^{T_{n_j+1}}\left(k-T_1-\cdots-T_{n_j}+\frac{\beta_{n_j}}{1-\beta_{n_j}}\right)
$$

$$
+ \beta_{n_k+1}^{r_k}\beta_{n_k}^{T_{n_k}}...\beta_{n_j+2}^{T_{n_j+2}}
$$

$$
\cdot\left(\beta_{n_j+1}^{r_j}\left(-(k-T_1-\cdots-T_{n_j}-1)-\frac{1}{1-\beta_{n_j+1}}\right)\right.
$$

$$
\left.+\left((k-T_1-T_{n_j}-r_j)+\frac{\beta_{n_j+1}}{1-\beta_{n_j+1}}\right)\right).
$$

On the right hand side, the first $n_j$ terms are non-positive since $\beta_1 \leq \beta_2 \leq \ldots \leq \beta_n$. Therefore,

$$\sum_{i=1}^{j}(k-i)b_{k,i} \leq \beta_{n_k+1}^{r_k}\beta_{n_k}^{T_{n_k}}\ldots\beta_{n_j+1}^{T_{n_j+1}}\left(k - T_1 - \cdots - T_{n_j} + \frac{\beta_{n_j}}{1-\beta_{n_j}}\right)$$
$$+ \beta_{n_k+1}^{r_k}\beta_{n_k}^{T_{n_k}}\ldots\beta_{n_j+2}^{T_{n_j+2}}$$
$$\left(\beta_{n_j+1}^{r_j}\left(-(k - T_1 - \cdots - T_{n_j} - 1) - \frac{1}{1-\beta_{n_j+1}}\right)\right.$$
$$\left.+ \left((k - T_1 - T_{n_j} - r_j) + \frac{\beta_{n_j+1}}{1-\beta_{n_j+1}}\right)\right).$$

By applying $\beta_{n_j+1}^{r_j} \geq \beta_{n_j+1}^{T_{n_j+1}}$ and $k - T_1 - \cdots - T_{n_j} - 1 = k - (j - r_j) - 1 \geq 0$ (since $j \leq k - 1$), we arrive at

$$\sum_{i=1}^{j}(k-i)b_{k,i} \leq \beta_{n_k+1}^{r_k}\beta_{n_k}^{T_{n_k}}\ldots\beta_{n_j+1}^{T_{n_j+1}}\left(k - T_1 - \cdots - T_{n_j} + \frac{\beta_{n_j}}{1-\beta_{n_j}}\right)$$
$$+ \beta_{n_k+1}^{r_k}\beta_{n_k}^{T_{n_k}}\ldots\beta_{n_j+2}^{T_{n_j+2}}$$
$$\left(\beta_{n_j+1}^{T_{n_j+1}}\left(-(k - T_1 - \cdots - T_{n_j} - 1) - \frac{1}{1-\beta_{n_j+1}}\right)\right.$$
$$\left.+ \left((k - T_1 - T_{n_j} - r_j) + \frac{\beta_{n_j+1}}{1-\beta_{n_j+1}}\right)\right)$$
$$\leq \beta_{n_k+1}^{r_k}\beta_{n_k}^{T_{n_k}}\ldots\beta_{n_j+1}^{T_{n_j+1}}\left(\frac{\beta_{n_j}}{1-\beta_{n_j}} + 1 - \frac{1}{1-\beta_{n_j+1}}\right)$$
$$+ \beta_{n_k+1}^{r_k}\beta_{n_k}^{T_{n_k}}\ldots\beta_{n_j+2}^{T_{n_j+2}}\left(k - T_1 - \cdots - T_{n_j} - r_j + \frac{\beta_{n_j+1}}{1-\beta_{n_j+1}}\right)$$
$$\leq \beta_{n_k+1}^{r_k}\beta_{n_k}^{T_{n_k}}\ldots\beta_{n_j+2}^{T_{n_j+2}}\left(k - T_1 - \cdots - T_{n_j} - r_j + \frac{\beta_{n_j+1}}{1-\beta_{n_j+1}}\right)$$
$$= \beta_{n_k+1}^{r_k}\beta_{n_k}^{T_{n_k}}\ldots\beta_{n_j+2}^{T_{n_j+2}}\left(k - j + \frac{\beta_{n_j+1}}{1-\beta_{n_j+1}}\right).$$

Now let us consider two cases: $r_k > 0$ and $r_k = 0$.

1. $r_k > 0$.

   In this case, we apply $\beta_1 \leq \ldots \leq \beta_n$ to get

   $$\sum_{i=1}^{j}(k-i)b_{k,i}$$
   $$\leq \beta_{n_k+1}^{r_k+T_{n_k}+\cdots+T_{n_j+2}}\left(k - j + \frac{\beta_{n_k+1}}{1-\beta_{n_k+1}}\right).$$

   Notice that

   $$r_k + T_{n_k} + \cdots + T_{n_j+2} = (T_1 + \cdots + T_{n_k} + r_k)$$
   $$- (T_1 + \cdots + T_{n_j} + T_{n_j+1})$$
   $$\leq (T_1 + \cdots + T_{n_k} + r_k)$$
   $$- (T_1 + \cdots + T_{n_j} + r_j)$$
   $$= k - j.$$

This tells us that

$$\sum_{i=1}^{j}(k-i)b_{k,i} \leq \beta_{n_k+1}^{k-j}\left(k-j+\frac{\beta_{n_k+1}}{1-\beta_{n_k+1}}\right).$$

Since $r_k > 0$, iteration $k$ is at the $(n_k+1)-$th stage, we have $\beta(k) = \beta_{n_k+1}$, and the above inequality is exactly what we want to show in (60).

2. $r_k = 0$

In this case, we apply $\beta_1 \leq ... \leq \beta_n$ to get

$$\sum_{i=1}^{j}(k-i)b_{k,i} \leq \beta_{n_k}^{T_{n_k}+\cdots+T_{n_j}+2}\left(k-j+\frac{\beta_{n_j+1}}{1-\beta_{n_j+1}}\right).$$

Notice that

$$\begin{aligned}
r_k + T_{n_k} + \cdots + T_{n_j+2} &= (T_1 + \cdots + T_{n_k} + r_k) \\
&\quad - (T_1 + \cdots + T_{n_j} + T_{n_j+1}) \\
&\leq (T_1 + \cdots + T_{n_k} + r_k) \\
&\quad - (T_1 + \cdots + T_{n_j} + r_j) \\
&= k - j.
\end{aligned}$$

This tells us that

$$\sum_{i=1}^{j}(k-i)b_{k,i} \leq \beta_{n_k}^{k-j}\left(k-j+\frac{\beta_{n_j+1}}{1-\beta_{n_j+1}}\right),$$

Since $r_k = 0$, we have $\beta(k) = \beta_{n_k}$ and by $j \leq k-1$ we deduce that $n_j \leq n_k - 1$ (Otherwise $j = T_1 + \cdots + T_{n_j} + r_j = T_1 + \cdots + T_{n_k} + r_j \geq T_1 + \cdots + T_{n_k} = k$). Therefore, we have

$$\sum_{i=1}^{j}(k-i)b_{k,i} \leq \beta_{n_k}^{k-j}\left(k-j+\frac{\beta_{n_k}}{1-\beta_{n_k}}\right),$$

which is exactly what we want to show in (60).

$\square$

# D  Main Theory for Multistage SGDM

In this section, we prove the main convergence theory of Multistage SGDM.

## D.1  Proof of Proposition 3

Proposition 3 is a generalization of Propositions 4 and 1 to the multistage case. Therefore, its proof is similar to those of Propositions 4 and 1.

First of all, by the smoothness of $f$ we have

$$\begin{aligned}
\mathbb{E}[f(z^{k+1})] &\leq \mathbb{E}[f(z^k)] + \mathbb{E}\langle\nabla f(z^k), z^{k+1}-z^k\rangle + \frac{L}{2}\mathbb{E}\|z^{k+1}-z^k\|^2 \\
&= \mathbb{E}[f(z^k)] + \mathbb{E}\langle\nabla f(z^k), -\alpha(k)\tilde{g}^k\rangle + \frac{L\alpha^2(k)}{2}\mathbb{E}\|\tilde{g}^k\|^2,
\end{aligned} \tag{62}$$

where we have applied Lemma 6 in the second step. Note that $\alpha(k)$ is the stepsize applied at the $k-$th iteration.

For the inner product term, we have

$$\mathbb{E}\langle\nabla f(z^k), -\alpha(k)\tilde{g}^k\rangle = \mathbb{E}\langle\nabla f(z^k), -\alpha(k)g^k\rangle,$$

which follows from the fact that $z^k$ is determined by the previous $k-1$ random samples $\zeta^1, \zeta^2, ...\zeta^{k-1}$, which is independent of $\zeta^k$, and $\mathbb{E}_{\zeta^k}[\tilde{g}^k] = g^k$.

As a result, we can write

$$
\begin{aligned}
\mathbb{E}\langle\nabla f(z^k), -\alpha(k)\tilde{g}^k\rangle &= \mathbb{E}\langle\nabla f(z^k) - g^k, -\alpha(k)g^k\rangle - \alpha(k)\,\mathbb{E}\,\|g^k\|^2 \\
&\leq \alpha(k)\frac{\rho_{0,k}}{2}L^2\,\mathbb{E}[\|z^k - x^k\|^2] + \alpha(k)\frac{1}{2\rho_{0,k}}\,\mathbb{E}[\|g^k\|^2] - \alpha(k)\,\mathbb{E}[\|g^k\|^2],
\end{aligned}
\tag{63}
$$

where $\rho_{0,k} > 0$ can be any positive constant.

Combining (62) and (63) gives

$$
\begin{aligned}
\mathbb{E}[f(z^{k+1})] \leq{}& \mathbb{E}[f(z^k)] + \alpha(k)\frac{\rho_{0,k}}{2}L^2\,\mathbb{E}[\|z^k - x^k\|^2] \\
&+ \left(\alpha(k)\frac{1}{2\rho_{0,k}} - \alpha(k)\right)\mathbb{E}[\|g^k\|^2] + \frac{L\alpha^2(k)}{2}\,\mathbb{E}[\|\tilde{g}^k\|^2]
\end{aligned}
$$

By (8) we know that $z^k - x^k = -A_1 m^{k-1}$, which leads to

$$
\begin{aligned}
\mathbb{E}[f(z^{k+1})] \leq{}& \mathbb{E}[f(z^k)] + \alpha(k)\frac{\rho_{0,k}}{2}L^2 A_1^2\,\mathbb{E}[\|m^{k-1}\|^2] \\
&+ \left(\alpha(k)\frac{1}{2\rho_{0,k}} - \alpha(k)\right)\mathbb{E}[\|g^k\|^2] + \frac{L\alpha^2(k)}{2}(\sigma^2 + \mathbb{E}[\|g^k\|^2]).
\end{aligned}
$$

Therefore, we have

$$
\begin{aligned}
\mathbb{E}[L^{k+1} - L^k] \leq{}& \alpha(k)\frac{\rho_{0,k}}{2}L^2 A_1^2\,\mathbb{E}[\|m^{k-1}\|^2] \\
&+ \left(\alpha(k)\frac{1}{2\rho_{0,k}} - \alpha(k) + \frac{L\alpha^2(k)}{2}\right)\mathbb{E}[\|g^k\|^2] + \frac{L\alpha^2(k)}{2}\sigma^2 \\
&+ c_1\alpha^2(k)\,\mathbb{E}[\|m^k\|^2] \\
&+ \sum_{i=1}^{k-1}(c_{i+1} - c_i)\,\mathbb{E}[\|x^{k+1-i} - x^{k-i}\|^2] \\
\leq{}& \alpha(k)\frac{\rho_{0,k}}{2}L^2 A_1^2\left(2\mathbb{E}[\|m^{k-1} - \sum_{i=1}^{k-1}b_{k-1,i}g^i\|^2] + 2\mathbb{E}[\|\sum_{i=1}^{k-1}b_{k-1,i}g^i\|^2]\right) \quad (64) \\
&+ \left(\alpha(k)\frac{1}{2\rho_{0,k}} - \alpha(k) + \frac{L\alpha^2(k)}{2}\right)\mathbb{E}[\|g^k\|^2] + \frac{L\alpha^2(k)}{2}\sigma^2 \\
&+ c_1\alpha^2(k)\left(2\mathbb{E}[\|m^k - \sum_{i=1}^{k}b_{k,i}g^i\|^2] + 2\mathbb{E}[\|\sum_{i=1}^{k}b_{k,i}g^i\|^2]\right) \\
&+ \sum_{i=1}^{k-1}(c_{i+1} - c_i)\,\mathbb{E}[\|x^{k+1-i} - x^{k-i}\|^2].
\end{aligned}
$$

On the other hand, we know that

$$
\begin{aligned}
\mathbb{E}[\|\frac{1}{1 - \prod_{i=1}^k \beta(i)}\sum_{i=1}^k b_{k,i}g^i\|^2] &= \mathbb{E}[\|\frac{1}{1 - \prod_{i=1}^k \beta(i)}\sum_{i=1}^k b_{k,i}g^i - g^k + g^k\|^2] \\
&\leq 2\,\mathbb{E}[\|g^k\|^2] + 2\,\mathbb{E}[\|\frac{1}{1 - \prod_{i=1}^k \beta(i)}\sum_{i=1}^k b_{k,i}g^i - g^k\|^2].
\end{aligned}
\tag{65}
$$

Furthermore,

$$\mathbb{E}[\|\frac{1}{1-\prod_{i=1}^{k}\beta(i)}\sum_{i=1}^{k}b_{k,i}g^i - g^k\|^2]$$

$$= \mathbb{E}[\|\frac{1}{1-\prod_{i=1}^{k}\beta(i)}\beta(k)\sum_{i=1}^{k-1}b_{k-1,i}g^i + \frac{1-\beta(k)}{1-\prod_{i=1}^{k}\beta(i)}g^k - g^k\|^2]$$

$$= \mathbb{E}[\|\frac{1}{1-\prod_{i=1}^{k}\beta(i)}\beta(k)\sum_{i=1}^{k-1}b_{k-1,i}g^i - \beta(k)\frac{1-\prod_{i=1}^{k-1}\beta(i)}{1-\prod_{i=1}^{k}\beta(i)}g^k\|^2] \quad (66)$$

$$= \beta^2(k)\left(\frac{1-\prod_{i=1}^{k-1}\beta(i)}{1-\prod_{i=1}^{k}\beta(i)}\right)^2 \mathbb{E}[\|\frac{1}{1-\prod_{i=1}^{k-1}\beta(i)}\sum_{i=1}^{k-1}b_{k-1,i}g^i - g^k\|^2].$$

Therefore, we have

$$\mathbb{E}[\|\frac{1}{1-\prod_{i=1}^{k-1}\beta(i)}\sum_{i=1}^{k-1}b_{k-1,i}g^i\|^2]$$

$$= \mathbb{E}[\|\frac{1}{1-\prod_{i=1}^{k-1}\beta(i)}\sum_{i=1}^{k-1}b_{k-1,i}g^i - g^k + g^k\|^2]$$

$$\leq 2\,\mathbb{E}[\|g^k\|^2] + 2\,\mathbb{E}[\|\frac{1}{1-\prod_{i=1}^{k-1}\beta(i)}\sum_{i=1}^{k-1}b_{k-1,i}g^i - g^k\|^2] \quad (67)$$

$$= 2\,\mathbb{E}[\|g^k\|^2] + 2\frac{1}{\beta^2(k)}\left(\frac{1-\prod_{i=1}^{k}\beta(i)}{1-\prod_{i=1}^{k-1}\beta(i)}\right)^2 \mathbb{E}[\|\frac{1}{1-\prod_{i=1}^{k}\beta(i)}\sum_{i=1}^{k}b_{k,i}g^i - g^k\|^2].$$

Plugging (65) and (67) into (64) gives us

$$\mathbb{E}[L^{k+1} - L^k]$$

$$\leq \left(-\alpha(k) + \alpha(k)\frac{1}{2\rho_{0,k}} + 2\alpha(k)\rho_{0,k}L^2 A_1^2 + \frac{L\alpha^2(k)}{2} + 4c_1\alpha^2(k)\right)\mathbb{E}[\|g^k\|^2]$$

$$+ \left(\alpha(k)\rho_{0,k}L^2 A_1^2 \mathbb{E}[\|m^{k-1} - \sum_{i=1}^{k-1}b_{k-1,i}g^i\|^2]) + \frac{1}{2}L\alpha^2(k)\sigma^2 + 2c_1\alpha^2(k)\mathbb{E}[\|m^k - \sum_{i=1}^{k}b_{k,i}g^i\|^2]\right)$$

$$+ \sum_{i=1}^{k-1}(c_{i+1} - c_i)\,\mathbb{E}[\|x^{k+1-i} - x^{k-i}\|^2]$$

$$+ 2\alpha(k)\rho_{0,k}L^2 A_1^2 \frac{1}{\beta^2(k)}\left(1 - \prod_{i=1}^{k}\beta(i)\right)^2 \mathbb{E}[\|\frac{1}{1-\prod_{i=1}^{k}\beta(i)}\sum_{i=1}^{k}b_{k,i}g^i - g^k\|^2]$$

$$+ 4c_1\alpha^2(k)\left(1 - \prod_{i=1}^{k}\beta(i)\right)^2 \mathbb{E}[\|\frac{1}{1-\prod_{i=1}^{k}\beta(i)}\sum_{i=1}^{k}b_{k,i}g^i - g^k\|^2] \quad (68)$$

In the rest of the proof, we will show that the sum of the last three terms in (68) is non-positive.

First, by Lemma 7 we know that

$$\mathbb{E}\,\|\frac{1}{1-\prod_{i=1}^{k}\beta(i)}\sum_{i=1}^{k}b_{k,i}g^i - g^k\|^2 \leq \sum_{i=1}^{k-1}a_{k,i}\,\mathbb{E}\,\|x^{i+1} - x^i\|^2,$$

where

$$a_{k,i} = \frac{L^2\beta^{k-i}(k)}{1-\prod_{i=1}^{k}\beta(i)}\left(k - i + \frac{\beta(k)}{1-\beta(k)}\right).$$

Or equivalently,

$$\mathbb{E}\left\|\frac{1}{1-\prod_{i=1}^{k}\beta(i)}\sum_{i=1}^{k}b_{k,i}g^i - g^k\right\|^2 \le \sum_{i=1}^{k-1}a_{k,k-i}\,\mathbb{E}\|x^{k+1-i}-x^{k-i}\|^2,$$

where

$$a_{k,k-i} = \frac{L^2\beta^i(k)}{1-\prod_{i=1}^{k}\beta(i)}\left(i+\frac{\beta(k)}{1-\beta(k)}\right).$$

Therefore, in order to make the sum of the last three terms of (68) to be non-positive, we need to enforce that

$$c_{i+1} \le c_i - \left(4c_1\alpha^2(k)\big(1-\prod_{i=1}^{k}\beta(i)\big)^2 + 2\alpha(k)\rho_{0,k}L^2A_1^2\frac{1}{\beta^2(k)}\big(1-\prod_{i=1}^{k}\beta(i)\big)^2\right)a_{k,k-i}$$

for all $i \ge 1$ and $k \ge 1$.

Since $1-\prod_{i=1}^{k}\beta(i) < 1$, $\beta_1 \le \beta(k) \le \beta_n$, and $\alpha_1 \le \alpha(k) \le \alpha_n$, we need to enforce the following for all $i \ge 1$:

$$c_{i+1} \le c_i - \left(4c_1\alpha_1^2 + 2\alpha(k)\rho_{0,k}L^2A_1^2\frac{1}{\beta_1^2}\right)\beta_n^i(i+\frac{\beta_n}{1-\beta_n})L^2.$$

Recall that $\frac{\alpha_i\beta_i}{1-\beta_i} \equiv A_1$ for all $n$ stages $i=1,2,...,n$. This gives us

$$c_{i+1} \le c_i - \left(4c_1\alpha_1^2 + 2\alpha(k)\rho_{0,k}L^2\frac{\alpha_1^2}{(1-\beta_1)^2}\right)\beta_n^i(i+\frac{\beta_n}{1-\beta_n})L^2.$$

Let us also set

$$\rho_{0,k} = \frac{1-\beta(k)}{2L\alpha(k)}. \tag{69}$$

Then, we need to enforce

$$c_{i+1} \le c_i - \left(4c_1\alpha_1^2 + 2\frac{1-\beta(k)}{2}L\frac{\alpha_1^2}{(1-\beta_1)^2}\right)\beta_n^i(i+\frac{\beta_n}{1-\beta_n})L^2.$$

Since $\beta_1 \le \beta(k)$, it suffices to enforce that

$$c_{i+1} = c_i - \left(4c_1\alpha_1^2 + L\frac{\alpha_1^2}{(1-\beta_1)}\right)\beta_n^i(i+\frac{\beta_n}{1-\beta_n})L^2. \tag{70}$$

Note that the equalities in (70) does not depend on $k$. In order for $c_i > 0$ for all $i \ge 1$, we can determine $c_1$ by

$$c_1 = \left(4c_1\alpha_1^2 + L\frac{\alpha_1^2}{(1-\beta_1)}\right)\sum_{i=1}^{\infty}\beta_n^i(i+\frac{\beta_n}{1-\beta_n})L^2.$$

Since

$$\sum_{i=1}^{j}i\beta_n^i = \frac{1}{1-\beta_n}\left(\frac{\beta_n(1-\beta_n^j)}{1-\beta_n}-j\beta_n^{j+1}\right),$$

we have $\sum_{i=1}^{\infty}i\beta_n^i = \frac{\beta_n}{(1-\beta_n)^2}$ and

$$c_1 = \left(4c_1\alpha_1^2 + L\frac{\alpha_1^2}{(1-\beta_1)}\right)\frac{\beta_n+\beta_n^2}{(1-\beta_n)^2}L^2.$$

This stipulates that

$$c_1 = \frac{\frac{\alpha_1^2}{(1-\beta_1)}\frac{\beta_n+\beta_n^2}{(1-\beta_n)^2}L^3}{1-4\alpha_1^2\frac{\beta_n+\beta_n^2}{(1-\beta_n)^2}L^2}. \tag{71}$$

Notice that $A_1 = \frac{1}{24\sqrt{2}L}$ and $\frac{1-\beta_1}{\beta_1} \leq 12\frac{1-\beta_n}{\sqrt{\beta_n+\beta_n^2}}$ ensures

$$4L^2\alpha_1^2\frac{\beta_n+\beta_n^2}{(1-\beta_n)^2} \leq \frac{1}{2}$$

and therefore

$$0 < c_1 \leq 2\frac{\alpha_1^2}{(1-\beta_1)}\frac{\beta_n+\beta_n^2}{(1-\beta_n)^2}L^3 \leq \frac{L}{4(1-\beta_1)}. \tag{72}$$

With the choices of $c_i$ in (70) and (71), the sum of the last three terms of (68) is non-positive. Therefore,

$$\mathbb{E}[L^{k+1} - L^k]$$
$$\leq \left(-\alpha(k) + \alpha(k)\frac{1}{2\rho_{0,k}} + 2\alpha(k)\rho_{0,k}L^2A_1^2 + \frac{L\alpha^2(k)}{2} + 4c_1\alpha^2(k)\right)\mathbb{E}[\|g^k\|^2]$$
$$+ \left(\alpha(k)\rho_{0,k}L^2A_1^2\mathbb{E}[\|m^{k-1} - \sum_{i=1}^{k-1}b_{k-1,i}g^i\|^2] + \frac{1}{2}L\alpha^2(k)\sigma^2 + 2c_1\alpha^2(k)\mathbb{E}[\|m^k - \sum_{i=1}^{k}b_{k,i}g^i\|^2]\right). \tag{73}$$

Taking $\rho_{0,k} = \frac{1-\beta(k)}{2L\alpha(k)}$ in (73) gives

$$\mathbb{E}[L^{k+1} - L^k]$$
$$\leq \left(-\alpha(k) + \frac{3-\beta(k)+2\beta^2(k)}{2(1-\beta(k))}L\alpha^2(k) + 4c_1\alpha^2(k)\right)\mathbb{E}[\|g^k\|^2]$$
$$+ \left(\frac{\beta^2(k)}{2(1-\beta(k))}L\alpha^2(k)\mathbb{E}[\|m^{k-1} - \sum_{i=1}^{k-1}b_{k-1,i}g^i\|^2] + \frac{1}{2}L\alpha^2(k)\sigma^2 + 2c_1\alpha^2(k)\mathbb{E}[\|m^k - \sum_{i=1}^{k}b_{k,i}g^i\|^2]\right).$$

Finally, by applying Lemma 4 and Lemma 5, we arrive at

$$\mathbb{E}[L^{k+1} - L^k]$$
$$\leq \left(-\alpha(k) + \frac{3-\beta(k)+2\beta^2(k)}{2(1-\beta(k))}L\alpha^2(k) + 4c_1\alpha^2(k)\right)\mathbb{E}[\|g^k\|^2]$$
$$+ \left(\frac{\beta^2(k)}{2}L\alpha^2(k)24\frac{\beta_1}{\sqrt{\beta_n+\beta_n^2}}\sigma^2 + \frac{1}{2}L\alpha^2(k)\sigma^2 + 4c_1(1-\beta_1)\alpha^2(k)\sigma^2\right).$$

### D.2   Proof of Theorem 3

From (73) we know that

$$\mathbb{E}[L^{k+1} - L^k] \leq -R_{1,k}\,\mathbb{E}[\|g^k\|^2] + R_{2,k}, \tag{74}$$

where

$$R_{1,k} = \alpha(k) - \alpha(k)\frac{1}{2\rho_{0,k}} - 2\alpha(k)\rho_{0,k}L^2A_1^2 - \frac{L\alpha^2(k)}{2} - 4c_1\alpha^2(k) \tag{75}$$

$$R_{2,k} = \alpha(k)\rho_{0,k}L^2A_1^2\mathbb{E}[\|m^{k-1} - \sum_{i=1}^{k-1}b_{k-1,i}g^i\|^2] + \frac{1}{2}L\alpha^2(k)\sigma^2 + 2c_1\alpha^2(k)\mathbb{E}[\|m^k - \sum_{i=1}^{k}b_{k,i}g^i\|^2]. \tag{76}$$

This immediately tells us that

$$L^1 \geq \mathbb{E}[L^1 - L^{k+1}] \geq \sum_{i=1}^{k}R_{1,i}\,\mathbb{E}[\|g^i\|^2] - \sum_{i=1}^{k}R_{2,i}, \tag{77}$$

In the rest the proof, we will bound $R_{1,i}$ and $R_{2,i}$ appropriately.

First, let us show that $R_{1,i} \geq \frac{\alpha(i)}{2}$ under $\rho_{0,i} = \frac{1-\beta(i)}{2L\alpha(i)}$ as in (69) and $\alpha(i) = \frac{A_1(1-\beta(i))}{\beta(i)} = \frac{1-\beta(i)}{24\sqrt{2}L\beta(i)}$.

From (72) we know that

$$c_1 \leq \frac{L}{4(1-\beta_1)}.$$

Therefore, in order for $R_{1,i} \geq \frac{\alpha(i)}{2}$, it suffices to have

$$\alpha(i)\frac{1}{2\rho_{0,i}} + 2\alpha(i)\rho_{0,i}L^2A_1^2 + \frac{L\alpha^2(i)}{2} + 4\frac{L}{4(1-\beta_1)}\alpha^2(i) \leq \frac{\alpha(i)}{2}. \tag{78}$$

By $\beta(i) \geq \beta_1 \geq \frac{1}{2}$ we know that

$$\alpha(i) = \frac{1-\beta(i)}{24\sqrt{2}L\beta(i)} \leq \frac{1}{2L}.$$

Therefore, $\frac{L\alpha^2(i)}{2} \leq \frac{\alpha(i)}{4}$. Furthermore, $\rho_{0,i} = \frac{1-\beta(i)}{2L\alpha(i)}$ yields

$$\alpha(i)\frac{1}{2\rho_{0,i}} + 2\alpha(i)\rho_{0,i}L^2A_1^2 + 4\frac{L}{4(1-\beta_1)}\alpha^2(i)$$

$$= \frac{L\alpha^2(i)}{1-\beta(i)} + \frac{\beta^2(i)L\alpha^2(i)}{(1-\beta(i))} + \frac{L}{(1-\beta_1)}\alpha^2(i)$$

$$\leq \frac{\alpha(i)}{12} + \frac{\alpha(i)}{12} + \frac{\alpha(i)}{12}$$

$$= \frac{\alpha(i)}{4},$$

where in the inequality above, we have applied

$$\alpha(i) = \frac{1-\beta(i)}{24\sqrt{2}L\beta(i)} \leq \frac{1-\beta(i)}{24L\frac{1}{2}} \leq \frac{1-\beta(i)}{12L},$$

$$\alpha(i) = \frac{1-\beta(i)}{24\sqrt{2}L\beta(i)} \leq \frac{1-\beta(i)}{12L\beta^2(i)},$$

$$\alpha(i) = \frac{1-\beta(i)}{24\sqrt{2}L\beta(i)} \leq \frac{1-\beta_1}{24L\beta_1} \leq \frac{1-\beta_1}{12L}.$$

Therefore, (78) is true and

$$R_{1,i} \geq \frac{\alpha(i)}{2}. \tag{79}$$

Now let us turn to $R_{2,i}$. By (76) and (72) we know that

$$R_{2,i} = \alpha(k)\rho_{0,i}L^2A_1^2\mathbb{E}[\|m^{i-1} - \sum_{j=1}^{i-1}b_{i-1,j}g^j\|^2] + \frac{1}{2}L\alpha^2(i)\sigma^2 + 2c_1\alpha^2(i)\mathbb{E}[\|m^i - \sum_{j=1}^{i}b_{i,j}g^j\|^2].$$

$$\leq \alpha(k)\rho_{0,i}L^2A_1^2\mathbb{E}[\|m^{i-1} - \sum_{j=1}^{i-1}b_{i-1,j}g^j\|^2] + \frac{1}{2}L\alpha^2(i)\sigma^2 + \frac{L}{2(1-\beta_1)}\alpha^2(i)\mathbb{E}[\|m^i - \sum_{j=1}^{i}b_{i,j}g^j\|^2].$$

Since $\rho_{0,i} = \frac{1-\beta(i)}{2L\alpha(i)}$ and $\frac{\alpha(i)\beta(i)}{1-\beta(i)} \equiv A_1$, we have

$$R_{2,i} \leq \frac{1}{2}L\alpha^2(i)\beta^2(i)\frac{1}{1-\beta(i)}\mathbb{E}[\|m^{i-1} - \sum_{j=1}^{i-1}b_{i-1,i}g^j\|^2] + \frac{1}{2}L\alpha^2(i)\sigma^2 + \frac{L}{2(1-\beta_1)}\alpha^2(i)\mathbb{E}[\|m^i - \sum_{j=1}^{i}b_{i,j}g^j\|^2].$$

By applying Lemmas 4 and 5, we further have

$$R_{2,i} \leq 12L\alpha^2(i)\beta^2(i)\frac{\beta_1}{\sqrt{\beta_n + \beta_n^2}}\sigma^2 + \frac{3}{2}L\alpha^2(i)\sigma^2. \tag{80}$$

By putting (79) and (80) into (77) with $k = T_1 + T_2 + \cdots + T_n$, we obtain

$$\sum_{l=1}^{n} \frac{\alpha_l}{2} \sum_{i=T_1+\cdots+T_{l-1}+1}^{T_1+\cdots+T_l} \mathbb{E}[\|g^i\|^2] \leq L^1 + \sum_{l=1}^{n} T_l \left( 12L\alpha_l^2\beta_l^2 \frac{\beta_1}{\sqrt{\beta_n + \beta_n^2}}\sigma^2 + \frac{3}{2}L\alpha_l^2\sigma^2 \right).$$

Dividing both sides by $\frac{1}{2}nA_2 \equiv \frac{1}{2}n\alpha_l T_l$ gives

$$\frac{1}{n}\sum_{l=1}^{n} \frac{1}{T_l} \sum_{i=T_1+\cdots+T_{l-1}+1}^{T_1+\cdots+T_l} \mathbb{E}[\|g^i\|^2]$$

$$\leq \frac{2\left(f(x^1) - f^*\right)}{nA_2} + \frac{1}{n}\sum_{l=1}^{n} \left( 24\beta_l^2 \frac{\beta_1}{\sqrt{\beta_n + \beta_n^2}}L\alpha_l\sigma^2 + 3L\alpha_l\sigma^2 \right)$$

$$= \mathcal{O}\left( \frac{f(x^1) - f^*}{nA_2} \right) + \mathcal{O}(\frac{1}{n}\sum_{l=1}^{n} L\alpha_l\sigma^2).$$

# E   Details of computational infrastructure

All experiments were performed on a computing server with Intel(R) Core(TM) i9-9940X CPU @ 3.30GHz and NVidia GeForce RTX 2080 P8. The weights of the neural networks are initialized by the default, random initialization routines.