[Reviews · NeurIPS 2020]

Review 1

Summary and Contributions: The paper presents a convergence analysis of the popular Stochastic Gradient Descent method with momentum (SGDM) and its multistage variant. The paper focus on two classes of smooth functions: non-convex and strongly convex. It was shown that SGDM converges as fast as SGD for the above classes of functions. The theoretical convergence is verified through preliminary numerical testing. ---- Post Rebuttal ---- I have read the authors' response and other reviews and decided to keep the overall score unchanged. I would strongly suggest the authors to improve the presentation of their work as I suggested in my original review, include missing references and highlight more the benefits of the proposed approach. Regarding the issue on Lemma 1 that R3 mentioned I trust the claim of the authors that they would be able to fix it. In the opposite scenario (if it turns out that the issue is not fixable) i expect that the authors will withdraw their submission as the whole theory depends on this result.

Strengths: The theoretical understanding of SGDM is very limited and the paper tries to close this gap. The proposed analysis is made through a novel Lyapunov function that take advantage of the reduced variance of the SGDM update. The analysis of multistage SGDM in also novel (to the best of my knowledge there is no previous analysis of the multistage variant).

Weaknesses: The ideas of the paper could be interesting however the paper loses some points in terms of presentation. Also some claims are not really justified. For example the title mentioned "An Improved Analysis" but it was never really explained in detail why the proposed analysis justifies the word "improved". There are some limitations of existing papers in the Intro but this should be more clear in the main contributions of the work. In line 74, the authors mentioned: "To the best of our knowledge, this is the first convergence (and acceleration) guarantee for SGDM in the multistage setting." How the acceleration is justified? I understand that in the initial stage of the algorithm one can use larger step-size and thus the method could be faster but it is not clear how one can claim accelerated rate. I think a table with a summary of convergence rates is needed where the results of previous works will be presented in comparison with the rates of the proposed analysis. This will make the presentation much clearer. Important: Normally the update of stochastic heavy ball method (SGD with momentum) has no $1-\beta$ before the gradient in update (2). See for example the update (3) in [2]. How is this different from the standard presentation of the SGS with heavy ball momentum: $x^{k+1}=x^k-\alpha g^k+\beta (x^k-x^{k-1}).$ ? The authors denote the Lyapunov function with $L^k$ which could be confused with the smoothness parameter $L.$ The presentation of Theorem 1 looks informal. One should state the properties of the function in a Theorem. In this case is f(x) is nonconvex and smooth. For Theorem 2 it was assumed that $k>k_0$. In my opinion this means that the rate is asymptotic (especially if it turns out that $k_0$ is large) and needs to be highlighted. In Remark 1 point 2 the following is mentioned: "assumes uniformly gradient of f". What this means? In the appendix, the proofs of Lemma 1 and Lemma 2 should change order. Also in line 314 it is mentioned that in the last step item 2 of Assumption 1 is used. Can you elaborate more on this? Why is that the case?

Correctness: I check the most important steps in the proofs and they seem correct. The experiments are adequate for a theoretical paper.

Clarity: See my previous comments. In general i believe that the paper could have been much better in terms of presentation and clarity of the main results.

Relation to Prior Work: Closely related references: [1] Loizou, Nicolas, and Peter Richtarik. "Momentum and stochastic momentum for stochastic gradient, Newton, proximal point and subspace descent methods." arXiv preprint arXiv:1712.09677 (2017). (the following is very recent paper/after ICML deadline. I am adding it here for the benefit of the authors) [2] Sebbouh, Othmane, Robert M. Gower, and Aaron Defazio. "On the convergence of the Stochastic Heavy Ball Method." arXiv preprint arXiv:2006.07867 (2020).

Reproducibility: Yes

Additional Feedback:


Review 2

Summary and Contributions: The paper provides an improved analysis of SGD with momentum (SGDM). To be specific, the authors introduce a new potential function and show that SGDM can converge as fast as SGD for smooth strongly-convex/nonconvex objectives. In addition, they establish the faster convergence of SGDM in the multistage setting.

Strengths: - The paper introduces a new potential function to analyze the convergence. This technique might be able to gain a great deal of attention in the optimization community, especially for those who are interested in understanding acceleration. - The paper presents the first convergence results for SGDM in the multistage setting.

Weaknesses: - For the final claims in the strongly convex setting, the authors use $z_k$ instead of $x_k$. However, $z_k$ is not the output of SGDM. None of the previous analyses establishes their convergence based on the auxiliary sequence $z_k$. Since $z_k$ is not a convex combination of $x_k$, I am curious about how it will influence our results.

Correctness: All the claims and methods are correct.

Clarity: Yes,

Relation to Prior Work: Yes

Reproducibility: Yes

Additional Feedback: It would be better if authors can explain how the convergence of auxiliary sequences $z_k$ can imply the convergence of $x_k$ output by SGDM. -- after rubuttal --- I checked the feedback from authors and agree the technical issues pointed out by other reviews can be solved. So I keep my score unchanged


Review 3

Summary and Contributions: Based on the authors' response after rebuttal, it is possible to fix the error. -------------------------------------------- Thanks for your feedback. After carefully reading them, I still have the concern on the proof of Lemma1. ----------------------------------------------------------------------- This paper analyzes the SGD with momentum under different settings via Lyapunov functions. For strongly convex and non-convex functions, the SGDM enjoys the same convergence of SGD. While, the multistage setting leads to acceleration.

Strengths: This work is based on sound assumptions and the empirical evaluation seems reasonable to me.

Weaknesses: - (7) requires that \alpha_i and \beta_i have to change at the same time, which makes the multistage setting less practical. - Since that all the theorems on the SGDM don't improve the convergence rate of SGD, the results seems not interesting.

Correctness: -The proof of Lemma 1 seems wrong. In Line 313, " Var (m_k ) = .... ||..\tilde{g}_i - g_i ||^2... g_i is also a random variable, so it should be \E [g_i]. So the second equality does not hold any more. I agree that momentum decreases the variance and increases the bias, so the proof might be fixable. - The latter proofs using Lemma 1 don't hold.

Clarity: - The structure of the paper is clear. - Yet some notations are not written in a professional way, for example, in "Assumption 2 & 4", the authors should indicate that \E and Var are conditional expectation and conditional variance wrt the past noises, as well as in the latter proofs. - Nit: It would be better to keep the notations consistent, such as $g_k$ and $\nabla f(x_k)$.

Relation to Prior Work: Yes.

Reproducibility: Yes

Additional Feedback:


Review 4

Summary and Contributions: after rebuttal: I have read the rebuttal and the other review's. I keep my score unchanged. More formal derivations saying that multistage SGDM can achieve epsilon accuracy faster than SGDM would improve understanding of this paper. I believe it is possible to do since multistage SGDM allows for the larger stepsizes at initial optimization stages. ------------------------------------------------------------------ The paper presents an improved theoretical analysis of SGD with momentum (SGDM) for non-convex and strongly convex functions, which matches convergence rate of SGD when the stepsize is small. Paper also proposes and analyses a new multi-stage SGD with momentum - a momentum SGD with the learning rate schedule, where the learning rate is dropped several times during training.

Strengths: Theoretical analysis is interesting and gives new insights about training SGD with momentum.

Weaknesses: - In the provided theoretical analysis, SGDM requires smaller stepsizes than standard SGD, which means that still in some cases SGD is faster, which was not discussed in the paper. - the theoretical comparison of multi-stage SGDM to standard SGDM is missing: is the rate of multi-stage SGDM better than the best rate of SGDM to reach fixed accuracy epsilon? - There is no experimental comparison to the SGD without momentum baseline.

Correctness: - I did not check all the proofs but the theoretical result seems correct. - Fig. 2 doesn't match with it's description on lines 207-209. - Fig. 2, the final performance of SGDM and multistage SGDM is the same, does that contradict the theory?

Clarity: mostly yes, missing references to MNIST dataset on line 194 & Pytorch and Tensorflow on line 23. - it is also a bit unclear if Lemma 1 is a new result or was already existent in the literature.

Relation to Prior Work: yes

Reproducibility: Yes

Additional Feedback: - it is unclear why is it required to increase momentum coefficient while dropping the learning rate in multi-stage SGDM? - in theorem 1 & 2 maybe remind that you use Assumption 1. - the choice of hyperparameters is not discussed: for the image classification task (Fig. 3), none of the algorithms achieve stoa performance, although STOA is possible to achieve with momentum SGD with decreasing learning rate schedule.

[Author Response · NeurIPS 2020]

R1.1 *...the title mentioned "An Improved Analysis"...should be more clear in the main contributions...* Great suggestion!
Our theory introduces a new Lyapunov function to analyze momentum SGD methods, it does not require uniformly
bounded gradients and leads to *the first convergence bound for Multistage SGDM*. Will explain this in the contributions.

R1.2 *Important...Normally stochastic heavy ball method has no* $1 - \beta$... The two are equivalent, as claimed in Page 1
footnote. Their update vectors $m^k = (1 - \beta) \sum_{i=1}^{k} \beta^{k-i} \tilde{g}^i$ and $m^k = \sum_{i=1}^{k} \beta^{k-i} \tilde{g}^i$ only differ by a scaling.

R1.3 *...How the acceleration is justified?...a table is needed...* At the initial stages, the sublinear terms in the convergence
bounds dominate, and Multistage SGDM allows for larger stepsizes, so it is faster during the initial stages. This may
not hold for final stages, although we also observe acceleration in the final stages in our numerical tests. We apologize
for the confusion and will clarify this in the revision! A table will be added to compare the convergence bounds.

R1.4 *...theorem 1 looks informal. For Theorem 2 it was assumed that $k > k_0$....* We will include Assumption 1 in Thm
1. For Thm 2, $k_0 = \lfloor \frac{\log 0.5}{\log \beta} \rfloor$ is a fixed constant. we will highlight this fact in the paper.

R1.5 *...it is mentioned that item 2 is used...Can you elaborate...?* In fact by item 2 we have $\mathbb{E}_{\zeta^i}[\tilde{g}^i] = g^i$, where
$\mathbb{E}_{\zeta^i}$ refers to taking expectation with respect to the minibatch $\zeta^i$ at the $i$th iteration. Therefore, when expanding
$\mathbb{E}[\| \sum_{i=1}^{k} \beta^{k-i}(\tilde{g}^i - g^i)\|^2]$, we have $\mathbb{E}_{\zeta^1}\mathbb{E}_{\zeta^2}...\mathbb{E}_{\zeta^k}[\langle \beta^{k-i}(\tilde{g}^i - \mathbb{E}_{\zeta^i}[\tilde{g}^i]), \beta^{k-j}(\tilde{g}^j - \mathbb{E}_{\zeta^j}[\tilde{g}^j])\rangle] = 0$ when $i \neq j$.

R1.6 *Closely related references...* We have read these interesting papers! Will cite and discuss them in our paper.

R2.1 *...Since $z_k$ is not a convex combination of $x_k$, I am curious about how it will influence our results.* Interesting
question! We do have similar convergence results for $\mathbb{E}[f(x^k)]$: since $x^k = (1 - \beta) \sum_{i=2}^{k} \beta^{k-i} z^i + \beta^{k-1} z^1$, from
which we can quickly get $\mathbb{E}[f(x^k) - f^*] = \mathcal{O}(r^k + \alpha \sigma^2)$, where $r = \max\{\beta, 1 - \alpha\mu\}$. Will add this in the revision.

R3.1 *(7) requires that $\alpha_i$ and $\beta_i$ have to change...which makes the multistage setting less practical.* We respectfully
disagree. Stagewise SGD with different stepsizes $\alpha_i$ is widely applied in practice. In addition to this, our Multistage
SGDM only needs to compute $\beta_i$ using (7), which costs little. The update vectors follow a very simple recursion. We
also demonstrated the effectiveness of Multistage SGDM in our numerical tests.

R3.2 *...all the theorems on the SGDM don't improve SGD, the results seems not interesting.* SGDM as well as its
multistage variants are known to work well in training DNNs such as ResNet and DenseNet. But, their convergence
properties remain largely unexplained. Our work narrows the gap between theory and practice. As pointed out by R1,
R2, and R5, our analysis brings new insights on the effect of momentum and improves previous results.

The convergence rate of SGDM under strong convexity (Thm 2) already matches the lower bound in Prop. 3 of
arXiv:1803.05591, so the worst-case guarantee cannot be improved. We believe that the same holds under nonconvexity.

R3.3 *The proof of Lemma 1 seems wrong..might be fixable.* Thanks for catching the glitch! It is fixable. We have the
same upper bound for $\mathbb{E}[\|m^k - (1 - \beta) \sum_{i=1}^{k} \beta^{k-i} g^i\|^2]$, and the rest of the proof only needs minor changes.

R3.4 *some notations...* Agreed! $\mathbb{E}$ and Var should be $\mathbb{E}_{\zeta^k}$ and $\text{Var}_{\zeta^k}$, respectively. We will also use $g^k$ throughout.

R5.1 *...SGDM requires smaller stepsizes...* Agreed. It is possible that SGD is faster in certain cases. Will discuss this.

R5.2 *...is the rate of multi-stage SGDM better than SGDM?* Thank you for pointing this out. Multistage SGDM is faster
during the initial stages (See R1.2). We will clarify this!.

R5.3 *... no experimental comparison to the SGD...* Thanks for the valuable suggestion! The comparison with SGD on
MNIST can be found in the figure below, where SGD has a similar performance as SGDM when $\alpha = 0.095$, and is
slightly slower when $\alpha = 0.66$. We will also add SGD in ResNet18 experiments.

*R5.4 Fig. 2 doesn't match with it's description...* We apologize
that Fig.2 used a wrong file due to an inadvertent overwriting. The
description in lines 207-209 corresponds to the figure on the left,
where we have also two added vanilla SGD curves as requested.
Multistage SGDM is faster in both the initial and final stages. The
curves can be reproduced by our submitted code.

R5.5 *...why is it required to increase momentum coefficient in Multistage SGDM?* The convergence theory for Multistage
SGDM requires the $\beta_i$ to satisfy eq (7). We agree that Multistage SGDM also works with a fixed momentum.

R5.6 *...none of the algorithms achieve SOTA performance...* We agree that using parameter choices with convergence
guarantees may not give the SOTA performance. However, we just find that Multistage SGDM can also achieve a test
accuracy of $93.0\%$ with 200 epochs. Will mentioned this.

[Meta-Review · NeurIPS 2020]

The paper studies the convergence of SGD with momentum, which is of strong research interest. It shows that SGD with momentum converges as fast as SGD for smooth strongly-convex/non-convex objectives, and faster in a multi-stage scenario with learning-rate decay. While the core contribution was liked by all reviewers, Reviewer 3 brought a serious issue in the proof of Lemma 1 to our attention, which forms the foundation for the main results. After the feedback and additional clarification by the authors and longer discussions, we share the impression with the authors that the issue can be fixed by replacing E[m^k] by v^k throughout the paper and adjusting minor constants. We expect trust the authors to perform these changes and should any issues remain, withdraw the paper. Additionally, we hope the detailed feedback with improvement suggestions from the 4 reviews will be implemented for the camera ready version.